# Antibiotic Stewardship in Treatment of Osteoarticular Infections Based on Local Epidemiology and Bacterial Growth Times

Pauline Vidal,[a] Eric Fourniols,[b] Helga Junot,[c] Cyril Meloni,[c] Alexandre Bleibtreu,[d]
Alexandra Aubry,[a,e] on behalf of the CRIOAC Pitié-Salpêtrière

[a]AP-HP, Laboratoire de Bactériologie-Hygiène, Sorbonne-Université, Hôpital Pitié-Salpêtrière, Paris, France
[b]AP-HP, Service de Chirurgie orthopédique, Hôpitaux Universitaires Pitié Salpêtrière-Charles Foix, Paris, France
[c]AP-HP, Pharmacie à usage intérieure, Hôpitaux Universitaires Pitié Salpêtrière-Charles Foix, Paris, France
[d]AP-HP, Service des Maladies infectieuses et Tropicales, Hôpitaux Universitaires Pitié Salpêtrière-Charles Foix, Paris, France
[e]Sorbonne Université, Inserm, U1135, Centre d'Immunologie et des Maladies Infectieuses, Paris, France

**ABSTRACT** Incubation for 14 days is recommended for the culture of microorganisms from osteoarticular infections (OAI), but there are no recommendations for postoperative antibiotic stewardship concerning empirical antimicrobial therapy (EAT), while prolonging broad-spectrum EAT results in adverse effects. The aim of this study was to describe the local OAI epidemiology with consideration of bacterial growth times to determine which antibiotic stewardship intervention should be implemented in cases of negative culture after 2 days of incubation. We performed a 1-year, single-center, noninterventional cohort study at the Pitié-Salpêtrière hospital OAI reference center. Samples were taken as part of the local standard of care protocol for adult patients who underwent surgery for OAI (native or device related) and received EAT (i.e., piperacillin-tazobactam plus daptomycin [PTD]) following surgery. The time to culture positivity was monitored daily. Overall, 147 patients were recruited, accounting for 151 episodes of OAI, including 112 device-related infections. Microbiological cultures were positive in 144 cases, including 42% polymicrobial infections. Overall, a definitive microbiological result was obtained within 48 h in 118 cases (78%) and within 5 days in 130 cases (86%). After 5 days, only Gram-positive bacteria were recovered, especially *Cutibacterium acnes*, *Staphylococcus* spp., and *Streptococcus* spp. Overall, 90% of culture-positive OAI were correctly treated with the locally established EAT. EAT guidance for OAI was in agreement with our local epidemiology. Our results supported antibiotic stewardship intervention consisting of stopping piperacillin-tazobactam treatment at day 5 in cases of negative culture.

**IMPORTANCE** Osteoarticular infections (OAI) remain challenging to diagnose and to treat. One of the issues concerns postoperative empirical antimicrobial therapy (EAT), which is usually a combination of broad-spectrum antibiotics. This EAT is maintained up to 2 weeks, until the availability of the microbiological results (identification and drug susceptibility testing of the microorganisms responsible for the OAI). Our results provide new data that will help to improve OAI management, especially EAT. Indeed, we have shown that antibiotic stewardship intervention consisting of stopping the antibiotic targeting Gram-negative bacteria included in the EAT could be implemented in cases where culture is negative after 5 days of incubation. The benefits of such an antibiotic stewardship plan include improved patient outcomes, reduced adverse events (including *Clostridioides difficile* infection), improvement in rates of susceptibilities to targeted antibiotics, and optimization of resource utilization across the continuum of care.

**KEYWORDS** osteoarticular infection, bacterial growth time, antibiotic stewardship, empirical antimicrobial therapy, epidemiology

Address correspondence to Alexandra Aubry, alexandra.aubry@sorbonne-universite.fr.

The authors declare no conflict of interest.

Osteoarticular infections (OAI), especially bone and prosthetic joint infections (PJI), cause significant morbidity and account for a substantial proportion of health care expenditures (1). To improve patients' rate of cure, it is highly valuable for empirical antimicrobial therapy (EAT) to be adapted as well as possible to the bacteria involved in an OAI. Guidelines provide recommendations for EAT for prosthetic joint and bone infections (2–4). This therapy should be started after surgery and is mostly based on a combination of a broad-spectrum beta-lactam, such as piperacillin-tazobactam or a third-generation cephalosporin, and an antibiotic effective against Gram-positive bacteria, such as vancomycin, daptomycin, or linezolid. Moreover, EAT should be adapted to the local microbial epidemiology (5–7).

The latter point illustrates how the microbiological diagnosis, including the identification of all bacteria involved in the OAI as well as drug susceptibility testing (DST) of all the bacteria involved in the OAI, is crucial for the management of OAI. Some bacteria involved in OAI are known to grow very slowly. They are usually described as belonging to the skin microbiota, such as coagulase-negative staphylococci (CoNS), *Cutibacterium acnes*, or anaerobic bacteria (8, 9). Some of these bacteria, especially *C. acnes*, are known to grow particularly slowly, requiring nearly 7 days to be detected in culture. For these reasons, 2-week microbiological cultures are recommended (10–13).

However, although European and American guidelines provide recommendations for the antibiotics to be included in EAT, none provides recommendations regarding the time at which EAT should be reevaluated, especially when cultures remain negative after 48 h (2–4). Nevertheless, establishing an antibiotic stewardship program aimed specifically at de-escalation could reduce adverse effects such as nephrotoxicity, prevent the emergence of antimicrobial-resistant bacteria and of *Clostridioides difficile* infection, and reduce the overall cost of treatment (14–17).

The aims of this cross-sectional study were (i) to describe the local epidemiology of OAI at Hôpitaux Universitaires Pitié Salpêtrière-Charles Foix, Paris, with consideration of bacterial growth times; (ii) to determine the reliability of local guidelines, especially the EAT; and (iii) to provide data to inform the stewardship program in order to define the right time for modifying the EAT for patients suspected of having OAI whose cultures remain negative after 2 days of incubation.

(A part of these results was presented previously at the RICAI, ECCMID, and CRIOAC congresses in 2021.)

## RESULTS

**Study population.** During 2020, 2,500 surgical procedures were performed in the orthopedic wards. Among those, 151 surgical procedures, corresponding to 147 patients, were followed by the initiation of an EAT for a suspected OAI. All those cases were included in this study (Table 1). Among the 151 cases, seven (corresponding to seven patients) were included only on the basis of strong confidence of the surgeon in the diagnosis; all other patients were included based on Musculoskeletal Infections Society (MSIS) criteria in addition to the surgeon's diagnosis.

Patients were mainly males ($n = 92$ [63%]), with a median age of 61 years (range, 18 to 98 years). One-third of the patients had underlying conditions known to be risk factors for OAI (Table 1) (18).

OAI was microbiologically confirmed in 144 cases (95%), whereas cultures remained negative in the seven remaining cases (5%). Of these seven cases, five were finally classified as "noninfected" according to the MSIS criteria, and two were classified as "inconclusive" according to the MSIS score of 5 (elevated serum C-reactive protein [CRP] levels with a perioperative sample containing an elevated number of leukocytes). Most of the cases were acute infections (68%). Sites of infection were spine ($n = 56$ [37%]), hip and femur ($n = 28$ [18%]), and knee ($n = 22$ [15%]) (Table 1). In most of the cases (74%), the infection involved an indwelling orthopedic device, including a joint prosthesis or an internal fixation device, whereas 9 OAI were related to a polytrauma. Four patients had been treated with antibiotics in the 15 days before the surgery, in two cases with positive and in two with negative cultures. These two latter cases were classified as "inconclusive" and "noninfected" according to

**TABLE 1** Patient characteristics, cases, and types of infection pertaining to 151 surgical procedures performed in the 147 patients included in this study[a]

| Characteristic | Value[b] for patients | | | P |
| --- | --- | --- | --- | --- |
| | Total | With monomicrobial infection | With polymicrobial infection | |
| Patients | 147[c] | 79 | 61 | |
| Median age (yr) | 61 (range, 18–98) | 59.5 (IQR, 44–73) | 59.5 (IQR, 41–74) | NS |
| Males | 92 (63) | 50 (53) | 39 (64) | NS |
| Females | 55 (37) | 29 (47) | 22 (36) | |
| | | | | |
| Underlying conditions[d] | 45 (31) | 22 (14) | 23 (15) | |
| Diabetes | 20 (14) | 9 (6) | 11 (7) | |
| Cancer and blood disorders | 15 (10) | 5 (3) | 10 (7) | |
| Immunosuppressive therapy | 10 (7) | 3 (2) | 7 (5) | |
| Active tobacco use | 10 (7) | 4 (3) | 6 (4) | |
| Kidney disease | 10 (7) | 4 (3) | 6 (4) | |
| | | | | |
| BMI | 25 (14–29) | 25 (17–28) | 25 (14–29) | NS |
| <18 | 8 (5) | 5 (3) | 3 (2) | |
| >30 | 26 (18) | 13 (8) | 13 (8) | |
| >18, <30 | 113 (77) | 56 (38) | 57 (39) | |
| | | | | |
| Location of OAI* | | | | 0.01 |
| Spine | 56 (37) | 33 (22) | 23 (15) | |
| Knee | 22 (15) | 15 (10) | 7 (5) | |
| Tibia/fibula | 18 (12) | 13 (9) | 5 (3) | |
| Hip | 17 (11) | 11 (7) | 6 (4) | |
| Foot | 14 (9) | 3 (2) | 11 (7) | |
| Femur | 11 (7) | 4 (3) | 6 (4) | |
| Ankle | 6 (4) | 3 (2) | 3 (2) | |
| Sacrum | 2 (1) | 0 (0) | 2 (1) | |
| Forearm | 2 (1) | 1 (<1) | 1 (<1) | |
| Sternum | 1 (<1) | 0 (0) | 1 (<1) | |
| Clavicle | 1 (<1) | 1 (<1) | 0 (0) | |
| Humerus | 1 (<1) | 0 (0) | 1 (<1) | |
| | | | | |
| Presentation of infection* | | | | |
| Positive culture | 144 | 80 (53) | 64 (42) | NS |
| Device related | 112 (74) | 63 (56) | 49 (44) | NS |
| Native articulation | 36 (24) | 16 (44) | 20 (56) | NS |
| Acute OAI | 94 (61) | 45 (33) | 48 (29) | NS |
| Chronic OAI | 50 (34) | 28 (18) | 21 (13) | NS |
| Hematogenous OAI | 7 (5) | 7 (5) | 0 (0) | NS |
| | | | | |
| Serum CRP > 10 mg/dL*[e] | 138 (92) | 77 (51) | 61 (40) | NS |
| Serum CRP, mean (range)*[e] | 93 (0–400) | 126 (0–400) | 83 (1.5–365) | NS |
| Sinus tract*[e] | 63 (41) | 35 (23) | 28 (19) | NS |
| Median time to positivity (h)[b] | 24 (24–47) | 24 (24–48) | 36 (24–48) | 0.005 |
| Antibiotic in the last 15 days | 4 (3) | | | |

[a]BMI, body mass index; OAI, osteoarticular infection.
[b]Values are number (percent) of patients unless otherwise noted by * when values are number of OAI.
[c]OAI was not confirmed microbiologically for 7 patients.
[d]Some patients have several underlying diseases.
[e]Criterion included in the MSIS definition.

the MSIS criteria (19) (Table 1). All except four patients received piperacillin-tazobactam and daptomycin as EAT; the remaining four patients received meropenem instead of piperacillin-tazobactam, since they were treated for a bacteremia due to a drug-resistant member of the *Enterobacterales* or were known to have had OAI with a resistant bacterium in the preceding 2 months.

**Microbiological epidemiology of OAI.** Among the 144 cases with a microbiological diagnosis, 80 (55%) were monomicrobial and 64 (45%) were polymicrobial (Table 1). Table 2 lists the causative microorganisms. The more frequently isolated organisms were *Staphylococcus aureus* (*n* = 58), including three methicillin-resistant *S. aureus* (MRSA) strains, CoNS (*n* = 57), and *Enterobacterales* (*n* = 54) (Table 2). Foot OAI were more frequently polymicrobial than OAI at

**TABLE 2** Epidemiology of osteoarticular infections, presented according to bacterial growth time

| Microorganism (no. of isolates)[a] | Number of isolates recovered according to the day of incubation[b]: | | | | | | | | | | Value for systematic subculture of broth |
| --- | --- | --- | --- | --- | --- | --- | --- | --- | --- | --- | --- |
| | D1 | D2 | D3 | D4 | D5 | D6 | D7 | D8 | D9 | D10 | |
| *Staphylococcus aureus* (58) | 38 | 18 | 1 | 0 | 0 | 0 | 0 | 0 | 0 | 0 | 1 |
| CoNS (57) | 13 | 24 | 12 | 0 | 2 | 2 | 0 | 1 | 0 | 3 | 0 |
| *Enterobacterales* (54) | 27 | 25 | 1 | 0 | 1 | 0 | 0 | 0 | 0 | 0 | 0 |
| *Enterococcus* spp. (18) | 6 | 12 | 0 | 0 | 0 | 0 | 0 | 0 | 0 | 0 | 0 |
| *Pseudomonas aeruginosa* (14) | 3 | 9 | 0 | 0 | 2 | 0 | 0 | 0 | 0 | 0 | 0 |
| *Cutibacterium acnes* (12) | 0 | 1 | 0 | 0 | 4 | 3 | 0 | 1 | 0 | 1 | 2 |
| Anaerobes (12) | 3 | 6 | 3 | 0 | 0 | 0 | 0 | 0 | 0 | 0 | 0 |
| *Streptococcus* spp. (11) | 5 | 4 | 1 | 0 | 0 | 0 | 0 | 0 | 0 | 1 | 0 |
| *Corynebacterium* spp. (10) | 1 | 8 | 1 | 0 | 0 | 0 | 0 | 0 | 0 | 0 | 0 |
| *Acinetobacter* spp. (2) | 1 | 1 | 0 | 0 | 0 | 0 | 0 | 0 | 0 | 0 | 0 |
| *Aeromonas hydrophila* (1) | 0 | 1 | 0 | 0 | 0 | 0 | 0 | 0 | 0 | 0 | 0 |
| *Stenotrophomonas maltophilia* (1) | 0 | 1 | 0 | 0 | 0 | 0 | 0 | 0 | 0 | 0 | 0 |
| *Pasteurella multocida* (1) | 1 | 0 | 0 | 0 | 0 | 0 | 0 | 0 | 0 | 0 | 0 |
| *Actinomyces odontolyticus* (1) | 0 | 1 | 0 | 0 | 0 | 0 | 0 | 0 | 0 | 0 | 0 |
| *Bacillus cereus* (1) | 1 | 0 | 0 | 0 | 0 | 0 | 0 | 0 | 0 | 0 | 0 |
| *Candida parapsilosis* (1) | 0 | 1 | 0 | 0 | 0 | 0 | 0 | 0 | 0 | 0 | 0 |
| *Candida albicans* (1) | 0 | 1 | 0 | 0 | 0 | 0 | 0 | 0 | 0 | 0 | 0 |
| No. of patients | 73 | 45 | 6 | 0 | 6 | 4 | 0 | 2 | 0 | 5 | 3 |

[a]CoNS, coagulase negative staphylococci.
[b]D, day.

other sites. Mono- or polymicrobial patterns differed among infection sites, especially for those occurring in fewer than 10 OAI cases ($P = 0.01$). There was no difference in terms of body mass index (BMI), age, and presence of material between cases of mono- and polymicrobial OAI (Table 1).

**Growth delay and time to obtain final microbiological results.** Almost all bacteria grew in the first 48 h after surgery (82%), whereas the median time to culture positivity was 24 h (interquartile range [IQR], 24 to 48 h) (Fig. 1). Overall, definitive microbiological results were obtained within 48 h in 118 cases (78%) and within 5 days in 130 cases (86%). After 5 days, only Gram-positive bacteria were observed in culture, which was liquid medium, only for 9 of the 14 corresponding OAI cases, including *C. acnes*, *Staphylococcus* spp., and *Streptococcus* spp. (Table 2). Seven cases remained negative (5%). The median time to positivity was significantly shorter for monomicrobial than polymicrobial OAI (24 h versus 36 h; $P < 0.01$) (Tables 3 and 4); however, when each bacterial species was considered separately, the time to observe growth and to reach a definitive diagnosis was not significantly different between monomicrobial and polymicrobial infections (Table 3).

The differences between growth delay and time to obtain final microbiological results were also not significant with respect to type of infection (acute or chronic) ($P = 0.52$), presence or absence of material ($P = 0.53$), or site of infection ($P = 0.1$) (Table 1).

**Drug susceptibility profiles and adequacy of EAT.** Regarding the DST results and the antibiotics used in EAT, the current EAT used in our setting was appropriate for 90% of the microbiologically proven OAI. Using imipenem instead of piperacillin-tazobactam would have increased the appropriateness of EAT for microbiologically proven OAI to 94.5%, whereas if piperacillin-tazobactam had been replaced by a third-generation cephalosporin, the appropriateness of EAT for microbiologically proven OAI would have decreased to 78% (Table 5). Indeed, 8 OAI cases among the 144 microbiologically proven OAI were documented as having at least one strain resistant to imipenem (*Pseudomonas aeruginosa* and/or *Stenotrophomonas maltophilia*), whereas 32 OAI cases among the 144 microbiologically proven OAI had at least one strain resistant to a third-generation cephalosporin (anaerobes, extended-spectrum-$\beta$-lactamase [ESBL]-producing *Enterobacterales*, *Pseudomonas aeruginosa*, *Stenotrophomonas maltophilia*, *Aeromonas hydrophila*, and/or *Acinetobacter* spp.).

In light of these results, an algorithm for managing OAI in patients receiving post-surgical EAT could be proposed (Fig. 2). In cases with positive cultures, the EAT could be adapted at any time to an appropriate therapy targeting the bacteria that have

Bacteria (number of isolates)

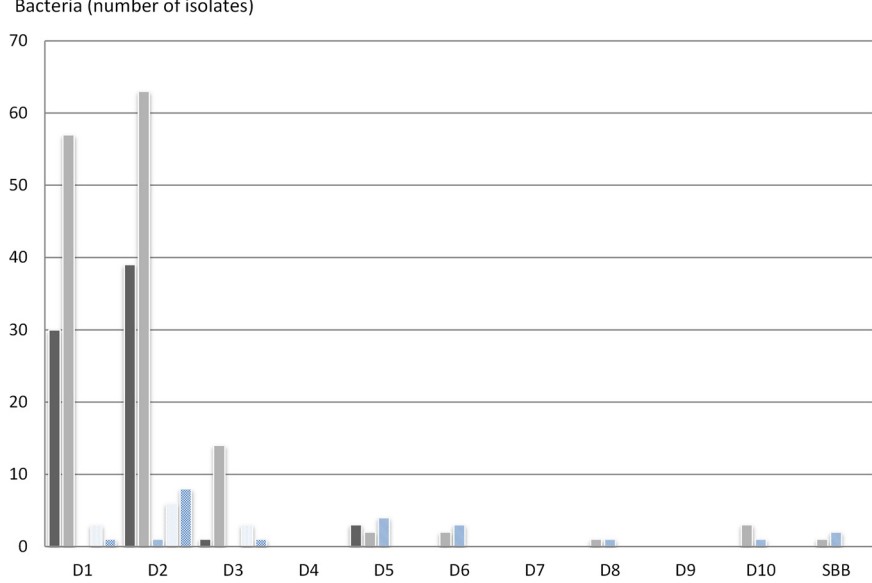

**FIG 1** Epidemiology of osteoarticular infections presented according to bacterial growth times.

been identified. In cases with negative cultures at day 5, the EAT would be stopped in the absence of MSIS criteria for OAI and/or high clinical suspicion of OAI, whereas it would be modified in cases that meet MSIS criteria for OAI and/or have high clinical suspicion of OAI. In the latter case, EAT would be modified by stopping piperacillin-tazobactam and using daptomycin as a monotherapy until definitive microbiological results are obtained. The proposed algorithm would be easy to implement in our setting, since all cases of OAI are reviewed twice a week by a multidisciplinary team to evaluate if a modification of patient care is required.

**TABLE 3** Epidemiology of osteoarticular infections, presented according to bacterial growth time and type of infection (monomicrobial or polymicrobial)[a]

| Microorganisms (no. of isolates) | Number of isolates recovered in monomicrobial or polymicrobial infection, according to the day of incubation: | | | | | | | | | | | | | | | | | | | | Value for systematic subculture in broth | |
|---|---|---|---|---|---|---|---|---|---|---|---|---|---|---|---|---|---|---|---|---|---|---|
| | D1 | | D2 | | D3 | | D4 | | D5 | | D6 | | D7 | | D8 | | D9 | | D10 | | | |
| | M | P | M | P | M | P | M | P | M | P | M | P | M | P | M | P | M | P | M | P | M | P |
| *Staphylococcus aureus* (58) | 31 | 7 | 8 | 10 | 0 | 1 | 0 | 0 | 0 | 0 | 0 | 0 | 0 | 0 | 0 | 0 | 0 | 0 | 0 | 0 | 1 | 0 |
| CoNS (57) | 4 | 9 | 6 | 18 | 1 | 11 | 0 | 0 | 0 | 2 | 0 | 2 | 0 | 1 | 0 | 0 | 0 | 1 | 2 | 0 | 0 | 0 |
| *Enterobacterales* (54) | 12 | 15 | 0 | 25 | 0 | 1 | 0 | 0 | 0 | 1 | 0 | 0 | 0 | 0 | 0 | 0 | 0 | 0 | 0 | 0 | 0 | 0 |
| *Enterococcus* spp. (18) | 1 | 5 | 0 | 12 | 0 | 0 | 0 | 0 | 0 | 0 | 0 | 0 | 0 | 0 | 0 | 0 | 0 | 0 | 0 | 0 | 0 | 0 |
| *Pseudomonas aeruginosa* (14) | 2 | 1 | 0 | 9 | 0 | 0 | 0 | 0 | 0 | 2 | 0 | 0 | 0 | 0 | 0 | 0 | 0 | 0 | 0 | 0 | 0 | 0 |
| *Cutibacterium acnes* (12) | 0 | 0 | 0 | 1 | 0 | 0 | 0 | 0 | 0 | 4 | 1 | 2 | 0 | 0 | 0 | 1 | 0 | 0 | 0 | 1 | 1 | 1 |
| *Anaerobes* (12) | 0 | 3 | 0 | 6 | 0 | 3 | 0 | 0 | 0 | 0 | 0 | 0 | 0 | 0 | 0 | 0 | 0 | 0 | 0 | 0 | 0 | 0 |
| *Streptococcus* spp. (11) | 4 | 1 | 2 | 2 | 0 | 1 | 0 | 0 | 0 | 0 | 0 | 0 | 0 | 0 | 0 | 0 | 0 | 1 | 0 | 0 | 0 | 0 |
| *Corynebacterium* spp. (10) | 1 | 0 | 0 | 8 | 0 | 1 | 0 | 0 | 0 | 0 | 0 | 0 | 0 | 0 | 0 | 0 | 0 | 0 | 0 | 0 | 0 | 0 |
| *Acinetobacter* spp. (2) | 0 | 1 | 0 | 1 | 0 | 0 | 0 | 0 | 0 | 0 | 0 | 0 | 0 | 0 | 0 | 0 | 0 | 0 | 0 | 0 | 0 | 0 |
| *Aeromonas hydrophila* (1) | 0 | 0 | 0 | 1 | 0 | 0 | 0 | 0 | 0 | 0 | 0 | 0 | 0 | 0 | 0 | 0 | 0 | 0 | 0 | 0 | 0 | 0 |
| *Stenotrophomonas maltophilia* (1) | 0 | 1 | 0 | 0 | 0 | 0 | 0 | 0 | 0 | 0 | 0 | 0 | 0 | 0 | 0 | 0 | 0 | 0 | 0 | 0 | 0 | 0 |
| *Pasteurella multocida* (1) | 1 | 0 | 0 | 0 | 0 | 0 | 0 | 0 | 0 | 0 | 0 | 0 | 0 | 0 | 0 | 0 | 0 | 0 | 0 | 0 | 0 | 0 |
| *Actinomyces odontolyticus* (1) | 0 | 0 | 0 | 1 | 0 | 0 | 0 | 0 | 0 | 0 | 0 | 0 | 0 | 0 | 0 | 0 | 0 | 0 | 0 | 0 | 0 | 0 |
| *Bacillus cereus* (1) | 1 | 0 | 0 | 0 | 0 | 0 | 0 | 0 | 0 | 0 | 0 | 0 | 0 | 0 | 0 | 0 | 0 | 0 | 0 | 0 | 0 | 0 |
| *Candida albicans* (1) | 0 | 1 | 0 | 0 | 0 | 0 | 0 | 0 | 0 | 0 | 0 | 0 | 0 | 0 | 0 | 0 | 0 | 0 | 0 | 0 | 0 | 0 |
| *Candida parapsilosis* (1) | 0 | 1 | 0 | 0 | 0 | 0 | 0 | 0 | 0 | 0 | 0 | 0 | 0 | 0 | 0 | 0 | 0 | 0 | 0 | 0 | 0 | 0 |
| No. of patients | 57 | 16 | 16 | 29 | 1 | 5 | 0 | 0 | 0 | 6 | 1 | 3 | 0 | 0 | 1 | 1 | 0 | 0 | 2 | 3 | 2 | 1 |

[a]D, day; M, monomicrobial infection; P, polymicrobial infection; CoNS, coagulase negative staphylococci.

**TABLE 4** Cumulative numbers of documented OAI according to the day of bacterial culture over the 151 surgical procedures

| Day | No. (%) of OAI | | |
| | Monomicrobial ($n = 80$) | Polymicrobial ($n = 64$) | Total ($n = 144$) |
| --- | --- | --- | --- |
| 1 | 57 (71) | 16 (25) | 73 (48) |
| 2 | 73 (91) | 45 (70) | 118 (78) |
| 3 | 74 (92) | 50 (78) | 124 (82) |
| 4 | 74 (92) | 50 (78) | 124 (82) |
| 5 | 74 (92) | 56 (87) | 130 (86) |
| 6 | 75 (94) | 59 (92) | 134 (89) |
| 7 | 75 (94) | 59 (92) | 134 (89) |
| 8 | 76 (95) | 60 (94) | 136 (90) |
| 9 | 76 (95) | 60 (94) | 136 (90) |
| 10 | 78 (98) | 63 (98) | 141 (93) |
| 13 | 80 (100) | 64 (100) | 144 (96) |

## DISCUSSION

The aim of this study was to describe the local epidemiology of OAI by focusing on the time to pathogen culture positivity in order to determine if an antibiotic stewardship intervention for de-escalation could be implemented. We designed a cohort study close to real-life clinical practice, since all patients with a suspected OAI for whom postoperative EAT was implemented were included. We have shown that EAT can be reevaluated and that a de-escalation may then be proposed at day 5 after surgery in cases in which cultures are still negative.

We also have shown that the antibiotic choice, driven by the epidemiology of the preceding year, to implement an EAT for patients suspected of OAI, was appropriate in most cases (90%). Our study highlighted that the definitive microbiological diagnosis was obtained in 48 h for 78% of OAI and in 5 days for 86%, while no Gram-negative bacteria (GNB) were recovered after 5 days of culture (Tables 2 and 3; Fig. 1), in both mono- and polymicrobial OAI. Prolonging cultures beyond day 5 made it possible to diagnose 10% more OAI cases, which justifies the strategy of using long incubations.

Polymicrobial OAI, which represent 42% of OAI in our setting, were documented later than monomicrobial OAI. However, as in other studies, only Gram-positive bacteria grew after 5 days (20–22) in both mono- and polymicrobial OAI. Only three GNB grew after the first 2 days, but still within 5 days after sampling. Apart from a small inoculum size, no factor explaining the late growth (at day 5) of these three GNB was identified (Table 2), especially no previous antibiotic exposure and no chronicity, confirming that a de-escalation may be

**TABLE 5** Susceptibility to piperacillin-tazobactam and cefepime of the microorganisms involved in OAI[a]

| Microorganism (no. of isolates) | No. (%) susceptible to: | | | |
| | Piperacillin-tazobactam | Cefepime | Third-generation cephalosporins | Imipenem |
| --- | --- | --- | --- | --- |
| *Staphylococcus aureus* (58) | 55 (95) | NR | NR | NR |
| CoNS (57) | 33 (58) | NR | NR | NR |
| *Enterobacteriaceae* (54) | 42 (78) | 46 (85) | 43 (79) | 54 (100) |
| *Enterococcus* spp. (18) | 16 (89) | NR | NR | 16 (89) |
| *Pseudomonas aeruginosa* (14) | 12 (86) | 12 (86) | 0 (0) | 7 (50) |
| *Cutibacterium acnes* (12) | 12 (100) | NR | NR | NR |
| Anaerobes (12) | 12 (100) | NR | 0 (0) | 12 (100) |
| *Streptococcus* spp. (11) | 11 (100) | 11 (100) | 11 (100) | 11 (100) |
| *Corynebacterium* spp. (10) | 0 (0) | 0 (0) | 0 (0) | 0 (0) |
| *Acinetobacter* spp. (2) | 1 (50) | 1 (50) | 0 (0) | 2 (100) |
| *Aeromonas hydrophila* (1) | 0 (0) | 1 (100) | 0 (0) | 0 (0) |
| *Stenotrophomonas maltophilia* (1) | 0 (0) | NR | 0 (0) | 0 (0) |
| *A. odontolyticus* (1) | 1 (100) | NR | 0 (0) | 1 (100) |
| *Pasteurella multocida* (1) | 1 (100) | NR | 1 (100) | 1 (100) |
| *Bacillus cereus* (1) | 0 (0) | 0 (0) | 0 (0) | 1 (100) |

[a]CoNS, coagulase-negative staphylococci; NR, not recommended for the treatment of infection due to that species.

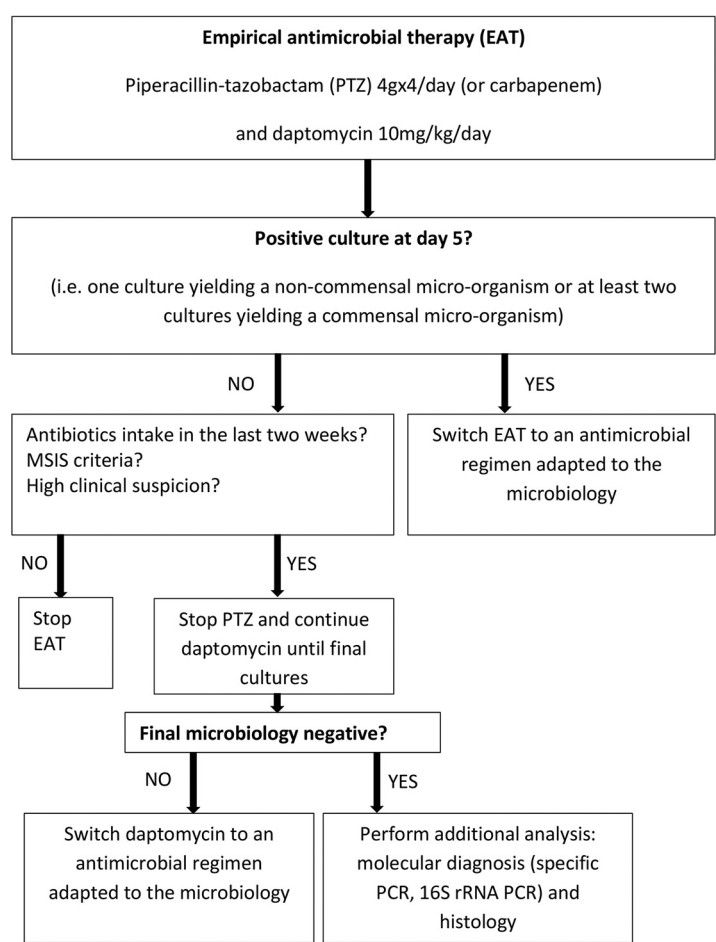

**FIG 2** Proposed algorithm for managing antibiotic administration for patients receiving postsurgical EAT.

proposed after day 5 (Tables 2 and 3; Fig. 2). As mentioned above, the Gram-positive bacteria growing after day 5 were *C. acnes*, followed by CoNS and *Streptococcus* spp. Similar findings were reported previously, e.g., in a French study aiming at estimating the right time to reevaluate EAT in hip and knee prosthetic joint infections, where more than 96% of the OAI-causing organisms were identified by day 5 and where the bacteria identified after 5 days were only Gram-positive bacteria, with only one case of *Enterobacterales* (20).

Therefore, we propose adapting EAT by continuing daptomycin as monotherapy in order to reduce patient exposure to broad-spectrum antibiotics as well as to their individual and collective side effects. Daptomycin is an appropriate anti-Gram-positive antibiotic in the postoperative EAT, since 42% of the bacteria recovered after 5 days were methicillin-resistant CoNS (Table 2).

In contrast to previous data, the epidemiology according to the type of infection (acute, chronic, or following specific surgical procedures) was not different between acute and chronic OAI (21% and 26% of OAI with commensal pathogens in acute and chronic infections, respectively) (Table 1) (8, 23–28). Some authors have suggested that it may be appropriate to tailor EAT to the clinical situation, since any noncommensal microorganisms could be involved in acute infections (e.g., *S. aureus*, enterococci, or GNB), whereas in late chronic infections, mostly CoNS and *C. acnes* are involved (23, 29). However, in our setting, piperacillin-tazobactam and daptomycin remained an appropriate choice for EAT, whatever the type of OAI (Tables 2 and 5).

Although we found a higher rate of polymicrobial infections (42%) than some studies (10% to 38% [30, 31]), our results are similar to those reported by other authors (20, 21, 23), suggesting that our conclusions may apply in different settings (Tables 2 and 3). The higher-than-usual

rate of polymicrobial OAI found in this study may be due to the specific recruitment in our setting, which is a large center for diabetic foot and polytrauma patients and patients with spine infections who are known to be likely to acquire polymicrobial infections (32–35). Indeed, 32% of the patients included in our study had internal osteosynthesis device infections of the spine. Finally, the fact that this study was conducted during the first waves of the SARS-CoV-2 pandemic may have added another bias in the type of patients who benefited from surgery (more polytrauma and emergency, especially spinal intervention, and less elective surgery). These factors combined with the local high-level expertise of our center, which is acknowledged as a reference center for OAI, also explain the high rate of confirmed positive microbiological diagnosis (96%).

In light of our results, EAT with piperacillin-tazobactam plus daptomycin (PTD) was appropriate in 90% of cases. To improve the rate of adequate EAT for OAI, we compared the number of OAI that would have been covered by a third-generation cephalosporin, cefepime, or imipenem instead of piperacillin-tazobactam (Table 5). Not surprisingly, EAT using cefepime or imipenem as the beta-lactam provide a higher appropriateness than the one containing a third-generation cephalosporin (92%, 94.5%, and 78%, respectively), especially because of the narrow spectrum of third-generation cephalosporin compared to cefepime and imipenem (36–38).

We would not recommend using third-generation cephalosporins in our EAT, since the decrease in the appropriateness of EAT for microbiologically proven OAI compared to the current recommendations is too important in our setting. We also would not recommend cefepime and imipenem in our EAT for several reasons: (i) the appropriateness of EAT for microbiologically proven OAI is not significantly different from what is observed with piperacillin-tazobactam; (ii) cefepime is not effective against anaerobes; which represent 6% of the OAI in our study but can vary from 2% to 30% according to the literature (23, 39, 40); and (iii) the local policy restricts the usage of carbapenems to patients who have bacteremia due to a resistant *Enterobacterales* or who have had OAI with a resistant bacterium in the preceding 2 months. However, it should be noted here that in each setting, the local epidemiology of OAI and patient characteristics should be assessed before an EAT is chosen.

According to Infectious Diseases Society of America (IDSA) guidelines, the benefits of antibiotic stewardship include improved patient outcomes, reduced adverse events (including *Clostridioides difficile* infection), improvement in rates of susceptibilities to targeted antibiotics, and optimization of resource utilization across the continuum of care (41). The algorithm for managing antibiotic administration in patients receiving postsurgical EAT that we propose (Fig. 2) is in line with those objectives. It is easy to implement, especially in settings where patient care is performed by a multidisciplinary team that includes microbiologists and infectious disease physicians, as in our setting. Our study has some limitations in that it was performed in a single center. Our hospital epidemiology may differ from that of other centers, underscoring the importance of knowing the local epidemiology in order to propose an appropriate EAT adaptation. Also, comparison of published data with those of the present study may be somewhat tricky, because ours was performed in France, whereas most similar studies were carried out in the United States or the United Kingdom (27, 28, 42–45), and some were performed more than a decade ago (27, 46, 47). Recent studies suggested that the microorganisms causing PJI can change over time or vary in different geographical areas (23, 48). Analyzing the local epidemiology on a regular basis is therefore appropriate.

Another limitation is that we included 4 patients who received meropenem and daptomycin, instead of piperacillin-tazobactam and daptomycin according to our local recommendations, because they carried multiresistant bacteria. However, given the small size of this population ($n = 4$), we consider that this limitation has minimal impact on observed results. Also, including 4 patients for 2 OAI each could have created a bias, as previous cultures were obtained and could have directed EAT. However, these 4 patients were treated with the standard EAT (i.e., PTD). We chose the same EAT knowing that strains involved in their first OAI were susceptible to PTD. Thus, we consider that the latter bias has minimal impact on our results.

**Conclusion.** Our results show that (i) the EAT used in our setting (i.e., PTD) is adapted to our local epidemiology and (ii) an antibiotic stewardship intervention consisting of stopping piperacillin-tazobactam at day 5 in cases with negative culture is appropriate. De-escalation to daptomycin monotherapy in such cases leads to a reduction in antibiotic exposure of the patient. Moreover, we have confirmed that prolonged culture on solid and in liquid media remains necessary for a bacteriological diagnosis in cases of OAI. Indeed, prolonged culture beyond 7 days should be maintained, as it allowed the diagnosis, in this study, of 10 (7%) additional cases, whereas the use of broth medium had a moderate impact, since it allowed diagnosis in only three (2%) additional cases. Finally, our study allowed the implementation of a decision algorithm for patients receiving postoperative EAT according to microbiological results (Fig. 2).

## MATERIALS AND METHODS

**Study design and data collection.** This single-center, noninterventional, cross-sectional study was set up, for a 1-year period (from 1 January 2020 to 31 December 2020), at the 1,500-bed University Hospital la Pitié-Salpêtrière, which is a reference center for complex OAI.

At least 3, and optimally 5 or 6, surgical samples were collected from each patient, as recommended by different guidelines (2, 4). In addition to the results of the culture of these perioperative samples, data on the following parameters were collected from the medical files: risk factors for OAI (i.e., obesity, active tobacco use, diabetes, cancer, and blood disorders [18]), preoperative puncture results (synovial white blood cell count, rate of polymorphonuclear and crystal deposition), inflammatory parameters (leukocyte count and C-reactive protein serum levels), perioperative aspect, and histopathological examination (19).

**Case definition.** In this study, a case was defined as a patient receiving a postoperative EAT for a suspected OAI, except facial-bone OAI. Patients received postoperative EAT if they met one of the following criteria: OAI suspicion according to the MSIS definition (19) and strong diagnostic confidence of the surgeon based on the clinical history, examination, inflammatory parameters, and imaging (19). All revision surgeries for a given patient were considered. In the case of a high degree of suspicion of OAI, surgeons collected 3 to 5 biopsy specimens for microbiological analysis, followed by the initiation of postoperative EAT based on a combination of piperacillin-tazobactam and daptomycin or carbapenem and daptomycin (for patients suspected of carrying multidrug-resistant bacteria), according to our hospital guidelines. These guidelines were established according to the epidemiology of the preceding year and European recommendations (49).

In this study, an acute OAI was defined as an OAI occurring within 3 months from the surgery, and a chronic OAI was defined as an OAI occurring more than 3 months after surgery according to previous work (19, 49, 50).

**Microbiological methods.** Cultures of tissue, liquid, or bone samples were performed following a standardized protocol. The corresponding samples were collected perioperatively from each patient in sterile vials. Specimens were homogenized and microscopically examined after Gram staining. The samples were immediately cultured on solid agar plates (containing 5% [vol/vol] sheep blood or chocolated horse blood) (Bio-Rad) and incubated for 14 days at 37°C simultaneously in aerobic, microaerophilic, and anaerobic atmospheres. Additionally, specimens were inoculated into brain heart infusion broth (Becton, Dickinson and Company [BD]) for enrichment, incubated at 37°C for 10 days, and then systematically subcultured for 72 h. The remaining samples were frozen at −40°C. Plates and broths were checked daily for microbial growth. All information concerning the day of positivity, the type of agar plates, and atmosphere as well as microbial identification was saved in a confidential data file.

Isolated colonies of bacteria or yeasts were identified by standard microbiological procedures, i.e., matrix-assisted laser desorption ionization–time-of-flight mass spectrometry (MALDI Biotyper, Bruker Daltonics, Leipzig, Germany). Antibiotic susceptibility testing was performed by the disk diffusion method as recommended (https://www.sfm-microbiologie.org/2021/04/23/casfm-avril-2021-v1-0/), while rapid testing for methicillin resistance on *Staphylococcus aureus* colonies was performed with the Alere PBP2a test.

The bacteriological results were considered positive if at least one culture yielded a strict pathogen (such as *S. aureus*, *Pseudomonas aeruginosa*, *Enterobacterales*, or anaerobes) or when at least two cultures yielded a commensal skin pathogen (such as CoNS or *C. acnes*) in the case of PJI (2). Time to positivity was defined as the time of growth to the first culture positivity of at least one sample in the case of noncommensal microorganisms such as *S. aureus*, *P. aeruginosa*, and *Enterobacterales* and as the time of growth from culture incubation to the first culture positivity in at least two samples in the case of commensal microorganisms such as CoNS and *C. acnes*. For polymicrobial cultures, time to positivity was defined as the time of growth of the last identified microorganism. The latter was considered positive if it was the sole culture yielding a strict pathogen (such as *S. aureus*, *Pseudomonas aeruginosa*, *Enterobacterales*, or anaerobes) or when it corresponded to a second positive culture yielding a commensal skin pathogen (such as CoNS or *C. acnes*) (2).

**Statistical analysis.** Results are expressed as number and percent for qualitative variables, and means and/or medians are used for quantitative variables. Pearson's $\chi^2$ test or Fisher's exact test was performed to compare qualitative variables, according to sample sizes. Wilcoxon's rank-sum test was used for quantitative variables. A $P$ value of $<0.05$ was considered statistically significant. Analysis was conducted using R software (51).

**Ethics.** The study protocol was approved by the national ethics committee for infectious diseases (CER-MIT 2020-0901). A information document was given to each patient.

## ACKNOWLEDGMENTS

We thank all personnel in charge of the OAI patients hospitalized at Pitié-Salpêtrière hospital as well as the technicians working at the laboratories. We also thank Ekkehard Collatz for English editing.

All authors declare no financial relationships in the previous 3 years with any organization that might have an interest in the submitted work and no other relationships or activities that could appear to have influenced the submitted work.

A.A., A.B., P.V.: designed the research. P.V.: data collection. A.A., P.V.: wrote the first draft of the paper. A.B., C.M., H.J., E.F.: contributed to the writing of the paper. All authors read and approved the final manuscript.

Members of the CRIOAC Pitié-Salpêtrière are as follows (in alphabetical order): Nicolas Barrut, Frédéric Clarençon, Georges Daas, Bruno Fautrel, Anne Fustier, Frédérique Gandjbakhch, Nagisa Godefroy, Frédéric Khiami, Jean Yves Lazennec, Maxime Marchant, Guillaume Mercy, Mihaela M. I. U, Gentiane Monsel, Quentin Monzani, Olivier Paccoud, Brigitte Rached, Vanessa Reubrecht, Jérôme Robert, and Noel Zahr.

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
