## [Reviewer comments · Microbiology Spectrum]

Microbiology Spectrum

Local epidemiology of osteoarticular infections with consideration of bacterial growth times provides useful data to inform antibiotic stewardship

Pauline Vidal, Eric Fourniols, Helga Junot, Cyril Meloni, Alexandre Bleibtreu, and Alexandra Aubry

Corresponding Author(s): Alexandra Aubry, AP-HP

Review Timeline:

Submission Date:	April 19, 2022
Editorial Decision:	June 22, 2022
Revision Received:	August 30, 2022
Editorial Decision:	September 28, 2022
Revision Received:	October 11, 2022
Accepted:	October 21, 2022

Editor: Tomefa Asempa

Reviewer(s): Disclosure of reviewer identity is with reference to reviewer comments included in decision letter(s). The following individuals involved in review of your submission have agreed to reveal their identity: Tristan Timbrook (Reviewer #2)

Transaction Report:

DOI: <https://doi.org/10.1128/spectrum.01430-22>

June 22, 2022

Dr. Alexandra Aubry
AP-HP
Sorbonne Université
91, boulevard de l'hôpital
Paris cedex 13 75013
France

Re: Spectrum01430-22 (Epidemiology of osteoarticular infections with consideration of bacterial growth times provides useful data to inform antibiotic stewardship)

Dear Dr. Alexandra Aubry:

Link Not Available

Sincerely,

Tomefa Asempa

Journals Department
Editor: Please revise thoroughly for grammatical errors and proof-read by native English speaker.

Reviewer comments:

Reviewer #1 (Comments for the Author):

This study describes the epidemiology of bone and joint infections at a Parisian university hospital. On the basis of this epidemiology and the time to positive culture antibiotic treatment could be adapted.

To my knowledge, the most recent recommendations on bone and joint infections (PJI) and osteoarticular infections (OAI) are those of the Musculoskeletal Infection Society (2018, (reference 18 in the paper)) and those of the European Bone & Joint Infection Society (2019, (1, 2)). These references should replace references 2 (2013) and 4 (2009).

In the Materials and methods chapter the authors mention that histopathological examination was carried out on surgical samples. It would be interesting to present the results of this exam. Did the histopathological results interfere with interpretation? Were they helpful?

Figure 1 should be omitted, there is no more information in this figure than in the text.

There is a problem in line 174: (24 days vs. 36 days, $p < 0.01$). Should it be 4 days vs 6 days?

Findings in Figure 2 do just repeat what is already shown in Tables 2 and 2bis; cf. Line 197: (Table 2, 2bis and Figure 2), line 206: (Table 2, Figure 2). This figure should be omitted.

In this study, all cases in which a post-operative empirical antibiotic treatment (EAT) was implemented were analyzed and 96% of these patients had a confirmed positive microbiological diagnosis. This seems an important number compared to other studies where positivity in empirically treated OAI patients is rather 60 to 70%. The authors should discuss this finding.

When EAT is justified in 96% of all patients, de-escalate after day 5 may concern only very few patients. In Table 2, there are 15 strains isolated after day 5 (5 of which were isolated on day 6) but it is not clear from how many patients these isolates were obtained. The number of patients concerned must be given. What is the correlation between the 255 isolates in Tables 2 and 2bis and the number of patients?

Were the isolates after day 5 cultured from solid or liquid media?

The comparison of piperacillin-tazobactam with cefepim in this study is problematic as cefepim is not recommended for an important number of bacterial species which are frequently isolated in OAI. The statement "If piperacillin-tazobactam were replaced by cefepim, the appropriateness of EAT would slightly increase (by 2%)" can only be true with regard to the local and temporal epidemiology here presented with a lack of efficacy on anaerobes, enterococci and most corynebacteria. This must be discussed precisely.

Minor remarks

- The authors should precise which were the companies providing materials (agar plates, etc).

- Line 146: seven negative cases (4%), line 173: seven negative cases (5%), is it 4 or 5%?

- Lines 200-201: "(90% of monomicrobial OAI were documented within two days vs 70% of polymicrobial OAI, with median times of 24h vs 36h, respectively; $p < 0.01$)"

90% in line 200 is 91% in Table 3. $P < 0.01$ should be $p = 0.005$ like in Table 1.

- "GNB" means Gram negative bacilli?

- There are many little typing errors which should be corrected ("and of Clostridioides", "OAI..", references 20 (initials are missing) and 37 (volume and pages are missing)).

(1) A. Signore, L.M. Sconfienza, O. Borens, A.W.J.M. Glaudemans, V. Casar-Pullicino, A. Trampuz, H. Winkler, O. Gheysens, F.M.H.M. Vanhoenacker, N. Petrosillo, P.C. Jutte. 2019.

Consensus document for the diagnosis of prosthetic joint infections: a joint paper by the EANM, EBJIS, and ESR (with ESCMID endorsement). European Journal of Nuclear Medicine and Molecular Imaging; 46: 971-88.

(2) A. W. J. M. Glaudemans, P.C. Jutte, M.A. Cataldo, V. Cassar-Pullicino, O. Gheysens, O.

Borens, A. Trampuz, K. Wörtler, N. Petrosillo, H. Winkler, A. Signore, L.M. Sconfienza. 2019.

Consensus document for the diagnosis of peripheral bone infection in adults: a joint paper by the EANM, EBJIS, and ESR (with ESCMID endorsement). European Journal of Nuclear Medicine and Molecular Imaging; 46: 957-70.

Reviewer #2 (Comments for the Author):

Overall: The present study is an epidemiological exploration into osteoarticular infections, evaluation of local treatment recommendations, and determination of potential culture-based timing for de-escalations. Beyond the epidemiology (trended with typical reported results), the authors noted an opportunity for de-escalations beyond 5 days for gram-negative coverage given the lack of yield beyond this time frame. Overall, these data are informative and interesting. I applaud the efforts of the authors in their work. Would recommend a variety of considerations before publishing.

Title: Akin to STROBE statement, would recommend making the title a little more explicit to the study considerations (e.g. study design....is this national data, local data, even a review article? Not determinable by current title).

Abstract

Lines 50-53: adults and peds? native and pji? Please specify.

Lines 55-60 and related in methods and discussion: Would consider excluding non-index encounter cases from antibiotic evaluation as previous cultures universally most predictive for present encounter etiology and thus directing therapy OR acknowledge inclusion in the limitations and then note the very small amt ($n=4$) and thus minimal impact on observed results

Line 62: "EAT for OAI was adapted to our local epidemiology" - clarify; do you mean EAT guidance was appropriate for the local epidemiology?

Methods

Lines 95-97: what risk factors were collected for OAI and where are these reported as results?

Lines 127-132: would consider clarifying if TTP is from culture collection or incubation.

Table 1: consider adding descriptive on MISS definitions; proportions of pji vs native sa

Table 1: how is the age range 18-98 yet stratification is 44 or 41 lower limit and 74 and 73 UL

Table 3 / Figure 1: would consider combining table 3 and figure 1

Reviewer #3 (Comments for the Author):

This was a well-written paper examining the epidemiology of OAIs and EAT. I have some concerns with this manuscript that will help improve its suitability for publication.

1. Introduction

- a. Consider altering the abbreviation of coagulase-negative Staphylococci to CoNS, as "CNS" is usually the default abbreviation for "central nervous system" instead and can be confusing for readers.
- b. The study aim is a bit vague, as "set up a most successful stewardship program" involves a large multitude of initiatives and interventions.

2. Materials and Method

- a. This sounds more like a cross-sectional study based on the study aim outlined in the introduction. Please utilize the STROBE checklist for cross-sectional studies to confirm that all parts are included.
- b. Please clarify "per-operative microbiological findings". What is meant by this phrase? How were the samples obtained? What kind of samples (skin, wound, bone, synovial fluid, abscess)?
- c. Please clarify "risk factors for OAI". What are these factors? Provide citations to support these factors.
- d. Please clarify if polymicrobial cultures were considered positive, as by the first sentence (starting line 125), they will not be considered "positive".
- e. Please update "Enterobacteriaceae" to "Enterobacterales" to coincide with the current nomenclature.
- f. In a cross-sectional study, there usually isn't a comparison between groups. Therefore, your use of inferential statistics does not make sense. Please clarify why inferential statistics were used and how that is applicable to the study aim.

3. Results

- a. Information is repeated in several sections. Please review to determine which section the duplicate information should actually live.
- b. Please provide a definition and citation for acute vs chronic infections. How many of these were trauma-related?
- c. Were devices removed? Any patients receive local antibiotics (e.g., bone cement, antibiotic spacers)?
- d. Please clarify "poorly represented sites". What does this mean?
- e. I believe there is a typo on units in line 175 for time to positivity. The reported units on this line say "days", but the rest of the paragraph is in hours. Please clarify.

4. Discussion/Conclusion

- a. There seems to be a significant amount of information duplicated both between tables, as well as between tables and figures. Please determine utility of all tables and figures and ensure that information is not duplicated within the text.
- b. This section should not be an introduction of new information (e.g., Figure 3). Please report this information in the results section.
- c. If the study aim is optimizing antimicrobial stewardship, there is no information in the results section on this. Additionally, only 2 sentences in the discussion section mention stewardship. As a study aim, 2 sentences is hardly appropriate. Further, please clarify that piperacillin/tazobactam is recommended to stop after 5 days post-operatively. Please revise.

Reviewer #4 (Comments for the Author):

Thank you for the opportunity to review the manuscript titled "Epidemiology of osteoarticular infections with consideration of bacterial growth times provides useful data to inform antimicrobial stewardship." In this manuscript, the authors sought to evaluate the patterns of microbiologic growth over time in order to provide guidance on what antimicrobial stewardship interventions could be appropriate for patients with osteoarticular infection and on empiric antimicrobial therapy. This is a practical and useful study that provides a road map for antimicrobial stewardship programs wanting to optimize management of osteoarticular infection. Programs can replicate the approach of looking at bacterial growth over time to inform local antimicrobial stewardship practices related to both type and timing of interventions in these infections. Strengths of the study include a contemporary cohort with a large number of diverse organisms, use of standard definitions (e.g. MSIS criteria), and straightforward statistical analysis. In addition, recruiting from a single institution, while limiting generalizability, also minimizes

variation in surgical and microbiological procedures. Major issues with the manuscript include lack of clarity in the methods (detailed below) and study population, as well as conclusions that exceed the scope of the results presented.

Major issues:

1. The case definition could be clarified. From my understanding, only patients who received post-operative empiric antibiotic therapy (EAT) were included. Four criteria are listed for receiving EAT: MSIS definition, diagnostic confidence of surgeon based on history, examination and inflammatory parameters. It would be extremely helpful to clarify if patients had to meet all criteria to receive EAT, any of the criteria, or specific combinations. When reporting the results, it would also be helpful to provide numbers of patients who met case definition based on MSIS definition, surgeon diagnosis, etc.
2. Were there any exclusion criteria? For example, patients who were suspected of OAI but did not have at least 3 surgical samples collected? Similarly, the MSIS criteria were used to define OAI, but were "inconclusive" or "not infected" cases were still included.
3. Could a patient be included in the study more than once, if they received post-operative EAT on more than one occasion?
4. A patient flow diagram would be helpful in understanding how the study sample relates to the institution's population. Perhaps # of patients referred for suspected OAI -> # of patients undergoing surgical management for suspected OAI -> # of patients with adequate sampling during surgery -> # of patients receiving post-operative EAT (i.e. the study sample).
5. The only statistical tests listed in methods (chi square and Fisher's exact tests) are for categorical/qualitative variables. What tests were used to compare continuous/quantitative variables (e.g. median time to positivity is measured in hours and should be compared with Wilcoxon rank-sum test).
6. The authors conclude that the proposed strategy to discontinue piperacillin/tazobactam at day 5 in negative culture cases is appropriate and safe. However, the authors have not evaluated the clinical outcomes of such a stewardship intervention; conclusions about safety are therefore inappropriate. The authors should consider a more limited conclusion (e.g. such a strategy is appropriate or justifiable based on local microbiology; do not discuss safety).
7. The authors conclude that prolonged culture is necessary to allow bacteriological diagnosis of OAI. However, the utility of bacteriologic diagnosis in these cases is unclear, as the authors propose therapy modifications based on culture data at day 5. The authors may wish to further reconcile these conclusions or explain the value of prolonged cultures beyond day 5, if therapy modification is made at that time.
8. In table 1, it is not clear why some variables are only presented overall (e.g. BMI) whereas others (e.g. age, location of OAI) are also compared between groups. Consider being consistent with between-group comparisons
9. Table 4 provides susceptibility of the collected pathogens to piperacillin/tazobactam (preferred agent) and cefepime (potential alternative). It would be helpful to also include meropenem (the alternative agent used at the author's institution) as well as at least one narrower spectrum agent (e.g. ceftriaxone). Comparison to a narrower regimen will support the conclusion that the preferred regimen of piperacillin/tazobactam + daptomycin is adapted to local epidemiology.

Minor issues:

1. The authors may wish to provide some explanation of existing prescribing practices at their institution. In particular, the routine use of daptomycin is not necessarily consistent with practice guidelines (e.g. daptomycin is not a "preferred therapy" in the IDSA prosthetic joint infection guidelines [Osmon et al. 2013]) or standards of care at other institutions. Why is vancomycin not used?
2. The methods indicate that "risk factors for OAI, pre-operative puncture results (synovial white blood cell count, rate of polymorphonuclear and crystal deposition), inflammatory parameters (leukocyte count and C-reactive protein serum levels), per-operative aspect, histopathological examination" were collected. However, that data are not clearly presented anywhere in the manuscript or tables. Perhaps expand Table 1 with some of this information, or create a new Table with these characteristics? I suspect this information was collected primarily to inform categorization according to MSIS criteria. If so, the authors could simply state that the listed data elements were collected in order to assess MSIS definition.
3. In the results, the authors discuss comparisons between type of infection (acute or chronic), but these types are not defined in the methods.
4. Figure 3: clarify if this was an existing algorithm prior to the results of this paper, or a proposed algorithm based on the results. If the latter, amending the title to "Proposed algorithm for taking charge..." would suffice. It is important to be clear on this so that readers can understand if the algorithm is supported by clinical outcomes data or only the bacterial epidemiology of osteoarticular infections.
5. Unless it is a journal standard, please report the actual result of non-significant p-values instead of "NS"
6. Time to positivity is listed as days in the text (line 175), should be hours.
7. Please review for consistency of abbreviations, spelling, typos, etc. (e.g. OAI is sometimes IOA [line 159, footnote to table 1], cefepime is sometimes cefepim, "resistant" is missing a letter on line 183 of text).

Reviewer #5 (Comments for the Author):

The premise of the study is interesting; however, I think the scope should be narrowed to, for example, prosthetic joint infections as many of your references are to this and not other types of osteoarticular infections such as diabetic foot infections. The epidemiological differences are important and either need to be addressed or the scope narrowed.

Please see the attached for more detailed comments.

Staff Comments:

Preparing Revision Guidelines

Please return the manuscript within 60 days; if you cannot complete the modification within this time period, please contact me. If you do not wish to modify the manuscript and prefer to submit it to another journal, please notify me of your decision immediately so that the manuscript may be formally withdrawn from consideration by Microbiology Spectrum.

This study attempts to describe the epidemiology of all osteoarticular infections at a single site in France and determine whether empiric antibiotic therapy following procedures can be deescalated prior to 14 days based on length to culture positivity.

Overall suggestions: This study groups all osteoarticular infections together; however, many references are made specifically to periprosthetic joint infections, including the guidelines referenced for diagnostic criteria and the choice of empiric antibiotic therapy. I suggest either focusing the study to periprosthetic joint infections. Alternatively, you need to include information on the other infections, such as the guidance for diagnosis and treatment of diabetic foot infections, hematogenous osteomyelitis, and vertebral osteomyelitis which tend to vary due to different epidemiological factors. Additionally, the surgical interventions for these infections can vary which will impact post-operative antibiotic therapy. For example, if complete source control is obtained in a diabetic foot infection as evidenced by clean margins on pathology, no further antibiotics would be needed even if cultures from the infection are positive.

Lines 44-48: This is unclear. The relationship between incubation time and post-operative antibiotic stewardship should be clarified. Also, the relationship of the aim to stewardship is unclear.

Line 51: Should “standard care” be “standard of care”?

Line 52-53: Clarify whether this is the EAT recommended at your institution or by other recommending bodies.

Lines 58-59: Are you trying to emphasize that these organisms may have been contaminants? If so, I recommend you clarify that. Alternatively, you could also choose to emphasize the characteristics/importance of bacteria grown in the first 5 days, especially since the following sentence and the conclusion seem to emphasize/advocate for focusing on bacteria grown within those first 5 days. Basically, clarify, the connection being drawn here one way of the other.

Lines 59-60: Given this article is couched in stewardship and a theme seems to be deescalating antibiotics, consider reframing this sentence to reflect that.

Introduction: The introduction should be stream-lined and focused. If your aim is to characterize OIA at your institution and look at growth times in terms of stewardship, focus on that.

Lines 69-72: The study covers different infections which have different guidance on empiric therapy. For example, some recommend covering for anaerobes of Gram negative organisms while others don't. If you're going to try to include all types of osteoarticular infections in one study, there needs to be more detail here. Otherwise, it may be better to simplify and focus on prosthetic joint infections.

Line 72: What guideline recommends gentamicin for empiric Gram positive coverage? If this is meant to indicate an antibiotic-impregnated spacer, that needs to be removed or clarified.

Lines 72-75: I think you're trying to make the point about the importance of local antibiograms. I agree with this point, but I think it can be made more clearly.

Lines 75-79: Again, I think this needs more detail given you've included so many types of osteoarticular infection, or narrow the scope of the paper to prosthetic joint infections. Some types of infections are

more likely to grow *C. acnes* for example, than others, so this becomes relevant to your point of how long to hold cultures.

Line 88: Rephrase this as the scope of the study didn't include setting up a stewardship initiative.

Lines 91-93: I suggest giving more background on existing stewardship and protocols for de-escalation. For example, are there infectious disease doctors who inform de-escalation? If not, I can understand the need for improved guidance for other practitioners. If so, this could be part of recommendations for improving de-escalation.

Line 95: What risk factors and for what kinds of OAIs?

Lines 101-105: The referenced criteria are for periprosthetic hip and knee infections; however, the study includes all osteoarticular infections, with no consideration for presence of prosthesis or location.

Lines 145-160: A lot of this should go under epidemiology. You could expand information on sex, age, underlying conditions, prior antibiotic exposure, and prior microbiological findings here.

Lines 167-168: Could this lack of a difference have been due to the small sample size, especially among polymicrobial infections and/or foot infections? It's unclear how much overlap there is between infection site and polymicrobial infections the way the data is presented. It may be clearer to just focus on prosthetic joint infections.

Lines 183-186: Table 4 looks at piperacillin-tazobactam and cefepime, not daptomycin. This needs to be corrected. Additionally, I think it's important to comment on daptomycin given that would have covered resistant Gram positives.

Lines 195-196: Were the antibiotics chosen based on your institution's prior antibiograms? If so, this isn't clear in the intro or methods. I would clarify this earlier on or rephrase this to indicate what empiric selection was actually based on (other guidelines for example).

Line 216: Were 42% methicillin resistant? This isn't clear from the table. If they're methicillin susceptible, you could narrow the antibiotic further.

Lines 217-219: I'm not sure this is a reasonable conclusion given you include all osteoarticular infections rather than focusing on one type, such as periprosthetic infections.

Line 223: I thought this was higher? 74%?

Line 233: I'm glad to see this point mentioned; however, I think it should come sooner. For example, this point could be included in Lines 107-108.

Lines 240-241: I'm sure this is true because piperacillin-tazobactam and daptomycin are broad spectrum. But, as this is meant to be a stewardship oriented study, this statement misses the point that narrower empiric therapy could be used in some cases.

Line 245: what is meant by "uneasy"? Please clarify this.

Table 1: Please add the number of females with infections, especially since you go on to give number of monomicrobial and polymicrobial infections.

Responses to reviewer's comments

Reviewer #1

This study describes the epidemiology of bone and joint infections at a Parisian university hospital. On the basis of this epidemiology and the time to positive culture antibiotic treatment could be adapted. To my knowledge, the most recent recommendations on bone and joint infections (PJI) and osteoarticular infections (OAI) are those of the Musculoskeletal Infection Society (2018, (reference 18 in the paper)) and those of the European Bone & Joint Infection Society (2019, (1, 2)). These references should replace references 2 (2013) and 4 (2009).

(1) A. Signore, L.M. Sconfienza, O. Borens, A.W.J.M. Glaudemans, V. Casar-Pullicino, A. Trampuz, H. Winkler, O. Gheysens, F.M.H.M. Vanhoenacker, N. Petrosillo, P.C. Jutte. 2019. Consensus document for the diagnosis of prosthetic joint infections: a joint paper by the EANM, EBJIS, and ESR (with ESCMID endorsement. European Journal of Nuclear Medicine and Molecular Imaging; 46: 971-88.

(2) A. W. J. M. Glaudemans, P.C.Jutte, M.A. Cataldo, V. Cassar-Pullicino, O. Gheysens, O. Borens, A. Trampuz, K. Wörtler, N. Petrosillo, H. Winkler, A. Signore, L.M. Sconfienza. 2019. Consensus document for the diagnosis of peripheral bone infection in adults: a joint paper by the EANM, EBJIS, and ESR (with ESCMID endorsement). European Journal of Nuclear Medicine and Molecular Imaging; 46: 957-70.

Answer: as suggested, the more recent references recommended by the reviewer were added.

In the Materials and methods chapter the authors mention that histopathological examination was carried out on surgical samples. It would be interesting to present the results of this exam. Did the histopathological results interfere with interpretation? Were they helpful?

Answer: histopathological examination was collected if performed. It is not systematically performed in our center, and, for example, any histopathological result had been performed for 7 patients with a negative culture.

Figure 1 should be omitted, there is no more information in this figure than in the text.

Answer: as suggested, we deleted Figure 1.

There is a problem in line 174: (24 days vs. 36 days, $p < 0.01$). Should it be 4 days vs 6 days?

Answer: we are sorry for the mistake, indeed, it should be 24 hours vs 36 hours. The text has been modified in the revised manuscript.

Findings in Figure 2 do just repeat what is already shown in Tables 2 and 2bis cf. Line 197: (Table 2, 2bis and Figure 2), line 206: (Table 2, Figure 2). This figure should be omitted.

Answer: we would prefer to keep the Figure 2 because we think that it gives a complementary visual of the results also presented in Tables 2 and 2bis. But, if editor and reviewers prefer to delete Figure 2, we will agree.

In this study, all cases in which a post-operative empirical antibiotic treatment (EAT) was implemented were analyzed and 96% of these patients had a confirmed positive microbiological diagnosis. This seems an important number compared to other studies

where positivity in empirically treated OAI patients is rather 60 to 70%. The authors should discuss this finding.

Answer: we agree it is a very high rate of confirmed positive microbiological diagnosis. It might be due to: (i) local high-level expertise of our center, as illustrated by the acknowledgement of our Center as a Reference Center for OAI, (ii) the peculiar profile of patients deserving surgery during the covid-19 pandemic as already said to discuss the high rate of polymicrobial infections. A sentence has been added into the discussion section in the revised manuscript (lines 259-270 in the revised manuscript with revised marked highlighted).

When EAT is justified in 96% of all patients, de-escalade after day 5 may concern only very few patients. In Table 2, there are 15 strains isolated after day 5 (5 of which were isolated on day 6) but it is not clear from how many patients these isolates were obtained. The number of patients concerned must be given. What is the correlation between the 255 isolates in Tables 2 and 2bis and the number of patients?

Answer: after day 5, the 15 strains isolated corresponded to 14 OAI (i.e. to 14 patients). We did add a line to precise the number of patients corresponding to the isolates in Tables 2 and 2bis in the revised version of the tables in order to clarify the point raised by the reviewer.

Were the isolates after day 5 cultured from solid or liquid media?

Answer: among the 14 IOA cases diagnosed after day 5, positive media were only liquid media for 9 cases. We added a sentence to include this information in the revised manuscript (line 197 in the revised manuscript with revised marked highlighted).

The comparison of piperacillin-tazobactam with cefepim in this study is problematic as cefepim is not recommended for an important number of bacterial species which are frequently isolated in OAI. The statement "If piperacillin-tazobactam were replaced by cefepim, the appropriateness of EAT would slightly increase (by 2%)" can only be true with regard to the local and temporal epidemiology here presented with a lack of efficacy on anaerobes, enterococci and most corynebacteria. This must be discussed precisely.

Answer: we agree that cefepime is not recommended for anaerobes, enterococci and most corynebacteria. However, whatever the beta-lactams used in the EAT we may propose, it will always be in combination with daptomycin which is active against gram-positive anaerobes, enterococci, and most corynebacteria. Therefore, the main target of the beta-lactams added in the EAT combination are gram-negative bacteria. In that purpose, cefepime is considered as an option in several teams (see references 1–3 here after).

The combination of cefepime and daptomycin may be ineffective against gram-negative anaerobes but more effective than piperacillin-tazobactam against some gram-negative bacteria producing beta-lactamases. Moreover, the percentage of OAI with both gram-negative and gram-positive anaerobes in our population represent only 6% of OAI and varies between 2 and 30% in the literature (see references 4-6 here after). As suggested by the reviewer, we modified the revised manuscript in order to better emphasize the usefulness of setting up an EAT strategy based on local epidemiology of OAI (line 273-280 in the revised manuscript with revised marked highlighted).

1. Triffault-Fillit C, Mabrut E, Corbin K, Braun E, Becker A, Goutelle S, et al. Tolerance and microbiological efficacy of cefepime or piperacillin/tazobactam in combination with vancomycin as empirical antimicrobial therapy of prosthetic joint infection: a propensity-matched cohort study. *J Antimicrob*

- Chemother. 2020;75(8):2299–306.
2. Giamarellou H. Fourth generation cephalosporins in the antimicrobial chemotherapy of surgical infections. *J Chemother* [Internet]. 1999 Dec;11(6):486–93.
 3. Robineau O, Talagrand-Reboulh E, Brunschweiler B, Jehl F, Beltrand E, Rousseau F, et al. Low prevalence of tissue detection of cefepime and daptomycin used as empirical treatment during revision for periprosthetic joint infections: results of a prospective multicenter study. *Eur J Clin Microbiol Infect Dis* [Internet]. 2021 Nov;40(11):2285–94.
 4. Gouliouris T, Aliyu SH, Brown NM. Spondylodiscitis: update on diagnosis and management. *J Antimicrob Chemother* [Internet]. 2010 Nov 1;65(Supplement 3):iii11–24.
 5. Liu Y, Su Y, Cui Z, Guo Y, Zhang W, Wu J. Clinical and microbiological features of anaerobic implant-related infection in 80 patients after orthopedic surgery. *Anaerobe* [Internet]. 2021 Oct;71:102413.
 6. Triffault-Fillit C, Ferry T, Laurent F, Pradat P, Dupieux C, Conrad A, et al. Microbiologic epidemiology depending on time to occurrence of prosthetic joint infection: a prospective cohort study. *Clin Microbiol Infect*. 2019;25(3):353–8.

Minor remarks

The authors should precise which were the companies providing materials (agar plates, etc).

Answer: as suggested, we added the companies providing materials in the revised manuscript.

Line 146: seven negative cases (4%), line 173: seven negative cases (5%), is it 4 or 5%?

Answer: the confusion come from the fact that the denominator was different in each sentence:

- Line 146, the denominator was the total number of the surgical procedures included in the study, i.e. 151: “Following 151 surgical procedures for suspected OAI, performed in the 147 patients of this study, OAI was microbiologically confirmed in 144 cases (96%), whereas cultures remained negative in the seven remaining cases (4%).”
- Line 173, the denominator was the total number of IOA cases, i.e. 144

Since the percentage was 4.6% in one case and 4.8% in the other, we modified the text and rounded up at 5% in order to avoid any further confusion.

Lines 200-201: "(90% of monomicrobial OAI were documented within two days vs 70% of polymicrobial OAI, with median times of 24h vs 36h, respectively; $p < 0.01$)" 90% in line 200 is 91% in Table 3. $P < 0.01$ should be $p = 0.005$ like in Table 1.

Answer: we apologize for this typing error which was corrected in the revised manuscript (91% as mentioned in Table 3 is the correct percentage).

"GNB" means Gram negative bacilli?

Answer: yes, "GNB" means gram-negative bacilli. The signification of GNB was added in the revised manuscript.

There are many little typing errors which should be corrected ("and ofClostridioides", "OAI..", references 20 (initials are missing) and 37 (volume and pages are missing).

Answer: we apologize for these typing errors that were corrected in the revised manuscript

Reviewer #2

Overall: The present study is an epidemiological exploration into osteoarticular infections, evaluation of local treatment recommendations, and determination of potential culture-

based timing for de-escalations. Beyond the epidemiology (trended with typical reported results), the authors noted an opportunity for de-escalations beyond 5 days for gram-negative coverage given the lack of yield beyond this time frame. Overall, these data are informative and interesting. I applaud the efforts of the authors in their work. Would recommend a variety of considerations before publishing.

Answer: we thank reviewer 2 for his/her very positive comments regarding our work.

Title: Akin to STROBE statement, would recommend making the title a little more explicit to the study considerations (e.g. study design....is this national data, local data, even a review article? Not determinable by current title).

Answer: as suggested, we modified the title to “Local epidemiology of osteoarticular infections with consideration of bacterial growth times provides useful data to inform antibiotic stewardship”.

Abstract.

Lines 50-53: adults and peds? native and pji? Please specify.

Answer: we added these data in the new manuscript (including in the abstract).

Lines 55-60 and related in methods and discussion: Would consider excluding non-index encounter cases from antibiotic evaluation as previous cultures universally most predictive for present encounter etiology and thus directing therapy OR acknowledge inclusion in the limitations and then note the very small amt (n=4) and thus minimal impact on observed results

Answer: we agree and modified the manuscript as suggested by the reviewer by acknowledging that the inclusion of this small population may be a limitation but with minimal impact on observed results. As 4 patients were included for 2 OAI each, if we would not have apply the same EAT for those 4 patients, it would have been bias. However, those 4 patients were treated with the same EAT (PTD) than other. We choose the same EAT knowing that strains involved in the first OAI were susceptible to PTD.

Moreover, 4 patients received meropenem and daptomycin because they carried multi-resistant bacteria. However, given the small size of this population (n=4), we consider this limitation has minimal impact on observed results. (lines 301-308 in the revised manuscript with revised marked highlighted).

Line 62: "EAT for OAI was adapted to our local epidemiology" - clarify; do you mean EAT guidance was appropriate for the local epidemiology?

Answer: yes, we did mean “EAT guidance was in agreement with the local epidemiology”, the text has been modified in the revised manuscript

Methods

Lines 95-97: what risk factors were collected for OAI and where are these reported as results?

Answer: risk factors collected were obesity, active tobacco use, diabetes, cancer and blood disorders, according to the literature (1) and the MSIS consensus conference (section I prevention). The manuscript has been modified in order to clarify this point made by reviewer 2 (line 102-103 in the revised manuscript with revised marked highlighted and Table 1).

- (1) Berbari EF, Hanssen AD, Duffy MC, Steckelberg JM, Ilstrup DM, Harmsen WS, et al. Risk factors for prosthetic joint infection: case-control study. Clin Infect Dis [Internet]. 1998 Nov;27(5):1247–54.

Lines 127-132: would consider clarifying if TTP is from culture collection or incubation.

Answer: as suggested, the revised manuscript has been modified in order to specify that TTP (Time To Positivity) was calculated from culture incubation, which is almost always the same day that culture collection (lines 144 in the revised manuscript with revised marked highlighted).

Table 1: consider adding descriptive on MISS definitions; proportions of pji vs native sa

Answer: as suggested, we added more descriptive on MISS definitions (proportions of pji vs native, CRP, sinus tract) in the revised table 1.

Table 1: how is the age range 18-98 yet stratification is 44 or 41 lower limit and 74 and 73 UL

Answer: we apologize for the missing information, indeed [18-98] corresponds to the range of age among patients, whereas for monomicrobial and polymicrobial columns, [44-73] and [41-74] correspond to IQR. The use of IQR was related to statistical analysis performed to compare monomicrobial and polymicrobial IOA. The table 1 has been modified in order to clarify the point made by reviewer2.

Table 3 / Figure 1: would consider combining table 3 and figure 1

Answer: as suggested by reviewer 1, Figure 1 has been deleted; therefore, there is no more possibility to combine both Table 3 and Figure 1.

Reviewer #3

This was a well-written paper examining the epidemiology of OAIs and EAT. I have some concerns with this manuscript that will help improve its suitability for publication.

Answer: we thank reviewer 3 for her/his very positive comments and for helping us to improve our manuscript.

1. Introduction

a. Consider altering the abbreviation of coagulase-negative Staphylococci to CoNS, as "CNS" is usually the default abbreviation for "central nervous system" instead and can be confusing for readers.

Answer: as suggested, we modified CNS for CoNS along the revised manuscript.

b. The study aim is a bit vague, as "set up a most successful stewardship program" involves a large multitude of initiatives and interventions.

Answer: as suggested, we narrowed and better specified the aim of the study in the abstract and the introduction sections of the revised manuscript (line 49 and lines 90-94 in the revised manuscript with revised marked highlighted).

2. Materials and Method:

a. This sounds more like a cross-sectional study based on the study aim outlined in the introduction. Please utilize the STROBE checklist for cross-sectional studies to confirm that all parts are included.

Answer: we checked the STROBE checklist for cross-sectional studies and confirmed that all parts are included in the manuscript.

b. Please clarify "per-operative microbiological findings". What is meant by this phrase? How were the samples obtained? What kind of samples (skin, wound, bone, synovial fluid, abscess)?

Answer: "per-operative findings" was modified to "results of the culture of per-operative samples" in order to clarify the sentence (line 101 in the revised manuscript with revised marked highlighted). Moreover, as written in the materials and methods section, "At least 3, optimally 5 or 6, surgical samples were collected from each patient, as recommended by different guidelines [2][4]." The nature of the samples depends on the tissues aspects, it can be bone, synovial fluid, abscess, etc.

c. Please clarify "risk factors for OAI". What are these factors? Provide citations to support these factors.

Answer: please see answer to reviewer 2 - question concerning "Methods, lines 95-97".

d. Please clarify if polymicrobial cultures were considered positive, as by the first sentence (starting line 125), they will not be considered "positive".

Answer: as suggested, we clarified the definition of the time to positivity for polymicrobial culture as following: the time of positivity is defined as the time of growth of the last identified micro-organism, by specifying "This latter is considered as positive if it was the sole culture yielding a strict pathogen (such as *S. aureus*, *Pseudomonas aeruginosa*, *Enterobacteriales* or anaerobes) or when it corresponded to a second positive culture yielding a commensal skin pathogen (such as CoNS or *C. acnes*) (2)".

e. Please update "Enterobacteriaceae" to "Enterobacteriales" to coincide with the current nomenclature.

Answer: as suggested, we modified "Enterobacteriaceae" to "Enterobacteriales" along the revised manuscript.

f. In a cross-sectional study, there usually isn't a comparison between groups. Therefore, your use of inferential statistics does not make sense. Please clarify why inferential statistics were used and how that is applicable to the study aim.

Answer: we thank the reviewer for raising this methodological point. We asked for the advice of colleagues who are biostatistician. Unfortunately, none of them had understood why reviewer 3 concluded that the statistics we used does not make sense. Therefore, we are afraid not being able to properly answer to this comment. Of course, we are willing to improve the manuscript if reviewer 3 can precise which statistics we should have applied.

3. Results:

a. Information is repeated in several sections. Please review to determine which section the duplicate information should actually live.

Answer: as suggested, we reviewed this section in order to avoid repetitions.

b. Please provide a definition and citation for acute vs chronic infections. How many of these were trauma-related?

Answer: we added the definition and citation for acute vs chronic infections in the material and methods section. As specified in the MSIS consensus conference, it is difficult to define acute and chronic infection. We used 3 months to discriminate between acute or chronic

infection as done in previous work (e.g. Rightmire E, Zurakowski D, Vrahas M. Acute infections after fracture repair: management with hardware in place. Clin Orthop Relat Res. 2008 Feb;466(2):466-72. doi: 10.1007/s11999-007-0053-y. Epub 2008 Jan 10. PMID: 18196433; PMCID: PMC25051191). See lines 118-120 in the revised manuscript with revised marked highlighted.

Nine of the cases were trauma related, this information has been added in the results section in the revised manuscript (see line 177 in the revised manuscript with revised marked highlighted).

c. Were devices removed? Any patients receive local antibiotics (e.g., bone cement, antibiotic spacers)?

Answer: devices were removed in 90% of devices related-OAI. In our center, surgeons do not use local antibiotics, except bone cement with antibiotic for mechanistic purpose (palacos[®] genta).

d. Please clarify "poorly represented sites". What does this mean?

Answer: as requested, we clarified the sentence and replaced "poorly represented sites" by "infected sites occurring in less than 10 OAI cases."

e. I believe there is a typo on units in line 175 for time to positivity. The reported units on this line say "days", but the rest of the paragraph is in hours. Please clarify.

Answer: the reviewer 3 is right, as also answered to reviewer 1 here above, the typing error has been corrected in the revised manuscript.

4. Discussion/Conclusion

a. There seems to be a significant amount of information duplicated both between tables, as well as between tables and figures. Please determine utility of all tables and figures and ensure that information is not duplicated within the text.

Answer: as already suggested by reviewer 1, we have deleted Figure 1.

b. This section should not be an introduction of new information (e.g., Figure 3). Please report this information in the results section.

Answer: as recommended, we reported this information in the results section in the revised manuscript, Figure 3 being Figure 2 in the updated manuscript since Figure 1 has been deleted (lines 211-220 in the revised manuscript with revised marked highlighted).

c. If the study aim is optimizing antimicrobial stewardship, there is no information in the results section on this. Additionally, only 2 sentences in the discussion section mention stewardship. As a study aim, 2 sentences is hardly appropriate. Further, please clarify that piperacillin/tazobactam is recommended to stop after 5 days post-operatively. Please revise.

Answer: as suggested by the reviewer, we added a paragraph to describe the stewardship that has been proposed in the results' section and also discuss it more extensively in the discussion section (lines 211-220 of the results section and lines 273-280 of the discussion section in the revised manuscript with revised marked highlighted).

Reviewer #4

Thank you for the opportunity to review the manuscript titled "Epidemiology of osteoarticular infections with consideration of bacterial growth times provides useful data to inform antimicrobial stewardship." In this manuscript, the authors sought to evaluate the patterns of microbiologic growth over time in order to provide guidance on what antimicrobial stewardship interventions could be appropriate for patients with osteoarticular infection and on empiric antimicrobial therapy. This is a practical and useful study that provides a road map for antimicrobial stewardship programs wanting to optimize management of osteoarticular infection. Programs can replicate the approach of looking at bacterial growth over time to inform local antimicrobial stewardship practices related to both type and timing of interventions in these infections. Strengths of the study include a contemporary cohort with a large number of diverse organisms, use of standard definitions (e.g. MSIS criteria), and straightforward statistical analysis. In addition, recruiting from a single institution, while limiting generalizability, also minimizes variation in surgical and microbiological procedures. Major issues with the manuscript include lack of clarity in the methods (detailed below) and study population, as well as conclusions that exceed the scope of the results presented.

Answer: we thank reviewer 4 for her/his very positive comments and for helping us to improve our manuscript.

Major issues:

1. The case definition could be clarified. From my understanding, only patients who received post-operative empiric antibiotic therapy (EAT) were included. Four criteria are listed for receiving EAT: MSIS definition, diagnostic confidence of surgeon based on history, examination and inflammatory parameters. It would be extremely helpful to clarify if patients had to meet all criteria to receive EAT, any of the criteria, or specific combinations. When reporting the results, it would also be helpful to provide numbers of patients who met case definition based on MSIS definition, surgeon diagnosis, etc.

Answer: case definition has been clarified in the material and method and in the results sections (lines 109-115 in the revised manuscript with revised marked highlighted). One of the criteria mentioned in the manuscript was sufficient to introduce EAT. Among the 151 cases, 7 cases were included based only on surgeon's conviction, whereas the latter 144 cases were included on both criteria (surgeons and MSIS criteria).

2. Were there any exclusion criteria? For example, patients who were suspected of OAI but did not have at least 3 surgical samples collected? Similarly, the MSIS criteria were used to define OAI, but were "inconclusive" or "not infected" cases were still included.

Answer: the only OAI excluded from our study were facial bones infection. Since the diagnostic confidence of surgeon was enough to include a patient whatever the MSIS classification. So, 2 patients having MSIS "inconclusive" criteria and 5 patients having MSIS "not infected" criteria were included. We added that information in the results section of the revised manuscript.

To clarify that point, we added the exclusion criteria in case definition in the materials and methods section (line 109 in the revised manuscript with revised marked highlighted).

3. Could a patient be included in the study more than once, if they received post-operative EAT on more than one occasion?

Answer: indeed, since all surgeries were considered if an EAT was prescribed, one patient could be included several times. It explains why 151 OAI cases were included in the study, corresponding to 147 patients (4 having 2 OAI). This has been clarified in the revised manuscript.

4. A patient flow diagram would be helpful in understanding how the study sample relates to the institution's population. Perhaps # of patients referred for suspected OAI -> # of patients undergoing surgical management for suspected OAI -> # of patients with adequate sampling during surgery -> # of patients receiving post-operative EAT (i.e. the study sample).

Answer: 2500 surgeries were performed in the orthopedic wards during the study period. Among those, 151 were followed by a post-operative EAT for a suspected OAI. Unfortunately, we do not have the information regarding the # of patients with adequate sampling during surgery, but we do know that those with post-operative EAT had at least 3 per-operative samples. Therefore, we cannot draw a patient flow diagram, but added the suggested valuable information to the revised manuscript (line 163 in the revised manuscript with revised marked highlighted).

5. The only statistical tests listed in methods (chi square and Fisher's exact tests) are for categorical/qualitative variables. What tests were used to compare continuous/quantitative variables (e.g. median time to positivity is measured in hours and should be compared with Wilcoxon rank-sum test).

Answer: as suggested, Wilcoxon rank-sum test has been used for quantitative variables in the revised manuscript.

6. The authors conclude that the proposed strategy to discontinue piperacillin/tazobactam at day 5 in negative culture cases is appropriate and safe. However, the authors have not evaluated the clinical outcomes of such a stewardship intervention; conclusions about safety are therefore inappropriate. The authors should consider a more limited conclusion (e.g. such a strategy is appropriate or justifiable based on local microbiology; do not discuss safety).

Answer: the reviewer 4 is right, since we have not evaluated the clinical outcomes, we should not have discussed safety. As recommended, we modified the conclusion section and replaced "safe" by "appropriate".

7. The authors conclude that prolonged culture is necessary to allow bacteriological diagnosis of OAI. However, the utility of bacteriologic diagnosis in these cases is unclear, as the authors propose therapy modifications based on culture data at day 5. The authors may wish to further reconcile these conclusions or explain the value of prolonged cultures beyond day 5, if therapy modification is made at that time.

Answer: we demonstrated that prolonged cultures allow to diagnose 10% more of the OAI. Our strategy enables to simplify the treatment by stopping piperacillin-tazobactam after day 5, whereas prolonged culture allows to narrow the antibiotic spectrum and replace daptomycin by orally given antibiotic which is a highly valuable for patients experiencing OAI (lines 232-233 in the revised manuscript with revised marked highlighted).

8. In table 1, it is not clear why some variables are only presented overall (e.g. BMI) whereas others (e.g. age, location of OAI) are also compared between groups. Consider being consistent with between-group comparisons

Answer: all variables were studied. But when statistical test was not significant, we wrote NS instead of a precise p value. We thought it might be more readable, but if editor and reviewers prefer to ass precise p values, we can add them to table 1.

Question: Table 4 provides susceptibility of the collected pathogens to piperacillin/tazobactam (preferred agent) and cefepime (potential alternative). It would be helpful to also include meropenem (the alternative agent used at the author's institution) as well as at least one narrower spectrum agent (e.g. ceftriaxone). Comparison to a narrower regimen will support the conclusion that the preferred regimen of piperacillin/tazobactam + daptomycin is adapted to local epidemiology.

Answer: we did not compare piperacillin-tazobactam to a 3rd-generation cephalosporin because it would be less effective than piperacillin-tazobactam given our local rate of *Pseudomonas aeruginosa* and other environment gram-negative bacilli. Regarding meropenem, the local policy restricted the usage of carbapenems (including meropenem) to patients experiencing bacteremia due to a resistant Enterobacterales or known to have had IOA with a resistant bacterium in the preceding two month. Therefore, we did not include meropenem (the alternative agent used at the author's institution) nor a 3rd-generation cephalosporin to our analysis (lines 273-280 in the revised manuscript with revised marked highlighted).

Minor issues:

1. The authors may wish to provide some explanation of existing prescribing practices at their institution. In particular, the routine use of daptomycin is not necessarily consistent with practice guidelines (e.g. daptomycin is not a "preferred therapy" in the IDSA prosthetic joint infection guidelines [Osmon et al. 2013]) or standards of care at other institutions. Why is vancomycin not used?

Answer: as other institutions, we choose to use daptomycin instead of vancomycin due to its easier way of administration and lower rate of adverse effects (e.g. reference here after: Joseph C, Robineau O, Titecat M, et al. Daptomycin versus Vancomycin as Post-Operative Empirical Antibiotic Treatment for Prosthetic Joint Infections: A Case-Control Study. *J Bone Jt Infect.* 2019 Mar 2;4(2):72-75. doi: 10.7150/jbji.22118. PMID: 31011511; PMCID: PMC6470651).

2. The methods indicate that "risk factors for OAI, pre-operative puncture results (synovial white blood cell count, rate of polymorphonuclear and crystal deposition), inflammatory parameters (leukocyte count and C-reactive protein serum levels), per-operative aspect, histopathological examination" were collected. However, that data are not clearly presented anywhere in the manuscript or tables. Perhaps expand Table 1 with some of this information, or create a new Table with these characteristics? I suspect this information was collected primarily to inform categorization according to MSIS criteria. If so, the authors could simply state that the listed data elements were collected in order to assess MSIS definition.

Answer: please see answer to reviewer 2's comment concerning "Methods, lines 95-97".

3. In the results, the authors discuss comparisons between type of infection (acute or chronic), but these types are not defined in the methods.

Answer: see answer to comment "3b" of the reviewer 3

4. Figure 3: clarify if this was an existing algorithm prior to the results of this paper, or a proposed algorithm based on the results. If the latter, amending the title to "Proposed algorithm for taking charge..." would suffice. It is important to be clear on this so that readers can understand if the algorithm is supported by clinical outcomes data or only the bacterial epidemiology of osteoarticular infections.

Answer: it is a proposed algorithm based on our results. As suggested, the title of Figure 3 (being Figure2 in the revised version of the manuscript) has been modified.

5. Unless it is a journal standard, please report the actual result of non-significant p-values instead of "NS"

Answer: when statistical tests were not significant we wrote NS instead of a precise p value. We thought it may be more readable, but if editor and reviewers prefer to ass precise p values, we can add them to table 1.

6. Time to positivity is listed as days in the text (line 175), should be hours.

Answer: we corrected the typing error, also highlighted by the reviewers 1 and 3.

7. Please review for consistency of abbreviations, spelling, typos, etc. (e.g. OAI is sometimes IOA [line 159, footnote to table 1], cefepime is sometimes cefepim, "resistant" is missing a letter on line 183 of text).

Answer: we apologize for these typing errors that were corrected in the revised manuscript.

Reviewer #5

This study attempts to describe the epidemiology of all osteoarticular infections at a single site in France and determine whether empiric antibiotic therapy following procedures can be deescalated prior to 14 days based on length to culture positivity.

Overall suggestions: This study groups all osteoarticular infections together; however, many references are made specifically to periprosthetic joint infections, including the guidelines referenced for diagnostic criteria and the choice of empiric antibiotic therapy. I suggest either focusing the study to periprosthetic joint infections. Alternatively, you need to include information on the other infections, such as the guidance for diagnosis and treatment of diabetic foot infections, hematogenous osteomyelitis, and vertebral osteomyelitis which tend to vary due to different epidemiological factors. Additionally, the surgical interventions for these infections can vary which will impact post-operative antibiotic therapy. For example, if complete source control is obtained in a diabetic foot infection as evidenced by clean margins on pathology, no further antibiotics would be needed even if cultures from the infection are positive.

Answer: none of the other 4 reviewers suggested to narrow the scope of the study to the prosthetic joint infections, and some even highlighted the usefulness of our "real life" based study. Therefore, we would prefer to keep the study design as it was initially planned since it may be valuable for readers working in the field of OAI.

Lines 44-48: This is unclear. The relationship between incubation time and post-operative antibiotic stewardship should be clarified. Also, the relationship of the aim to stewardship is unclear.

Answer: as suggested by reviewer 3 as well (see comment “1. introduction b”), we rephrased the aim of the study in the revised manuscript.

Line 51: Should “standard care” be “standard of care”?

Answer: we corrected this typing error and replaced “standard care” to “standard of care”.

Line 52-53: Clarify whether this is the EAT recommended at your institution or by other recommending bodies.

Answer: the EAT recommendations at our institution were established according to the epidemiology of the preceding year and European recommendations (18). This precision has been added in the revised manuscript.

Lines 58-59: Are you trying to emphasize that these organisms may have been contaminants? If so, I recommend you clarify that. Alternatively, you could also choose to emphasize the characteristics/importance of bacteria grown in the first 5 days, especially since the following sentence and the conclusion seem to emphasize/advocate for focusing on bacteria grown within those first 5 days. Basically, clarify, the connection being drawn here one way of the other.

Answer: indeed, our willing was to emphasize that after 5 days, micro-organisms recovered were only gram-positive bacteria which means that we could de escalate EAT with an antibiotic active only against gram-positive bacteria. All those bacteria were not considered as contaminants since they grew up in different samples and in both solid and liquid media (lines 232-233 in the revised manuscript with revised marked highlighted).

Lines 59-60: Given this article is couched in stewardship and a theme seems to be deescalating antibiotics, consider reframing this sentence to reflect that.

Answer: as also suggested by reviewer 3 (see comment “1. Introduction b”), we rephrased the aim of our study to reflect that in the revised manuscript.

Introduction The introduction should be stream-lined and focused. If your aim is to characterize OIA at your institution and look at growth times in terms of stewardship, focus on that.

Answer: as suggested, the introduction has been slightly modified in the revised manuscript.

Lines 69-72: The study covers different infections which have different guidance on empiric therapy. For example, some recommend covering for anaerobes of Gram negative organisms while others don’t. If you’re going to try to include all types of osteoarticular infections in one study, there needs to be more detail here. Otherwise, it may be better to simplify and focus on prosthetic joint infections.

Answer: as said here above we would prefer to keep the design of the study as initially planned.

Line 72: What guideline recommends gentamicin for empiric Gram positive coverage? If this is meant to indicate an antibiotic-impregnated spacer, that needs to be removed or clarified.

Answer: as suggested, gentamycin was removed from the text.

Lines 72-75: I think you’re trying to make the point about the importance of local antibiograms. I agree with this point, but I think it can be made more clearly.

Answer: as suggested, the text has been modified to make the importance of local antibiogram clearer in the revised manuscript (line 78 in the revised manuscript with revised marked highlighted).

Lines 75-79: Again, I think this needs more detail given you've included so many types of osteoarticular infection, or narrow the scope of the paper to prosthetic joint infections. Some types of infections are more likely to grow *C. acnes* for example, than others, so this becomes relevant to your point of how long to hold cultures.

Answer: as said here above, since the design of our study is appreciated by the other reviewers, we would prefer to keep the design of the study as initially planned.

Line 88: Reword this as the scope of the study didn't including setting up a stewardship initiative.

Answer: we reworded the aim of the study to clarify the scope of the study in the revised manuscript (line 49 and lines 90-94 in the revised manuscript with revised marked highlighted).

Lines 91-93: I suggest giving more background on existing stewardship and protocols for de-escalation. For example, are their infectious disease doctors who inform de-escalation? If not, I can understand the need for improved guidance for other practitioners. If so, this could be part of recommendations for improving de-escalation.

Answer: as suggested, we gave more information about antibiotic stewardship in the revised manuscript: each case is discussed twice a week with infectious disease physicians among other, to adapt EAT to the final bacteria involved in OAI or to start de-escalation in case of negative culture (lines 211-220 of the results section and lines 273-280 of the discussion section in the revised manuscript with revised marked highlighted).

Line 95: What risk factors and for what kinds of OAIs?

Answer: please see answer to reviewer 2's comment concerning "Methods, lines 95-97".

Lines 101-105: The referenced criteria are for periprosthetic hip and knee infections; however, the study includes all osteoarticular infections, with no consideration for presence of prosthesis or location.

Answer: as specified in the material and methods section, all cases in which a post-operative EAT was implemented were analyzed. Patients received post-operative EAT if they met only one of the following criteria: OAI suspicion according to the Musculoskeletal Infections Society (MSIS) definition and/or strong diagnostic confidence of the surgeon based on the clinical history helped with examination, inflammatory parameters and imaging. Cas definition has been rephrased to avoid any confusion (lines 108-113 in the revised manuscript with revised marked highlighted).

Lines 145-160: A lot of this should go under epidemiology. You could expand information on sex, age, underlying conditions, prior antibiotic exposure, and prior microbiological findings here.

Answer: the confusion comes from the word epidemiology leaved alone; we changed the name of this section to "microbiological epidemiology of OAI" instead of "epidemiology of OAI".

Lines 167-168: Could this lack of a difference have been due to the small sample size, especially among polymicrobial infections and/or foot infections? It's unclear how much overlap there is between infection site and polymicrobial infections the way the data is presented. It may be clearer to just focus on prosthetic joint infections.

Answer: indeed, the differences may be due to the small sample size. This limitation has been added to the discussion in the revised manuscript.

Lines 183-186: Table 4 looks at pip-taz and cefepime, not daptomycin. This needs to be corrected. Additionally, I think it's important to comment on daptomycin given that would have covered resistant Gram positives.

Answer: we do measure the susceptibility to daptomycin only for gram-positive bacteria resistant to piperacillin-tazobactam. Therefore, we do not have daptomycin susceptibility results for all the strains isolated in the frame of the study. Evaluating daptomycin susceptibility profile of all the strains isolated in our study was beyond the scope of the study. If needed for publication, we are willing to measure daptomycin MICs for the strains for which this data is missing.

Lines 195-196: Were the antibiotics chosen based on your institution's prior antibiograms? If so, this isn't clear in the intro or methods. I would clarify this earlier on or reword this to indicate what empiric selection was actually based on (other guidelines for example).

Answer: as suggested, we have indicated that the antibiotics chosen is based on our institution's OAI epidemiology and the European recommendations (18) in the materiel and method part of the revised manuscript.

Line 216: Were 42% methicillin resistant? This isn't clear from the table. If they're methicillin susceptible, you could narrow the antibiotic further.

Answer: the 42% referred to strains recovered after 5 days, all were methicillin resistant CoNS; but not all CoNS were methicillin resistant.

Lines 217-219: I'm not sure this is a reasonable conclusion given you include all osteoarticular infections rather than focusing on one type, such as periprosthetic infections.

Answer: none of the other 4 reviewers raised that point, since no argument in disfavor of our statement is given by the reviewer, we would be willing to keep that sentence.

Line 223: I though this was higher? 74%?

Answer: the confusion come from the fact that 74% correspond to device-related OAI and 32% to the device-related OAI in the spine area. To avoid confusion the text has been modified in the revised manuscript.

Line 233: I'm glad to see this point mentioned; however, I think it should come sooner. For example, this point could be included in Lines 107-108.

Answer: as suggested this paragraph has been moved earlier in the discussion section in the revised manuscript.

Lines 240-241: I'm sure this is true because pip-taz and daptomycin are broad spectrum. But, as this is meant to be a stewardship oriented study, this statement misses the point that narrower empiric therapy could be used in some cases.

Answer: we do not recommend C3G instead of piperacillin-tazobactam according to our microbiologic epidemiology. Please see our answer to comment “major issue – question” of the reviewer 4 (lines 273-280 in the revised manuscript with revised marked highlighted).

Line 245: what is meant by “uneasy”? Please clarify this.

Answer: we rephrased and replaced “uneasy” by “tricky”.

Table 1: Please add the number of females with infections, especially since you go on to give number of monomicrobial and polymicrobial infections.

Answer: as requested, we added the number of females with infections in the revised manuscript.

September 28, 2022

Dr. Alexandra Aubry
AP-HP
Sorbonne Université
91, boulevard de l'hôpital
Paris cedex 13 75013
France

Re: Spectrum01430-22R1 (Local epidemiology of osteoarticular infections with consideration of bacterial growth times provides useful data to inform antibiotic stewardship)

Dear Dr. Alexandra Aubry:

Thank you for submitting your manuscript to Microbiology Spectrum. As you will see your paper is very close to acceptance. Please modify the manuscript along the lines I have recommended. As these revisions are quite minor, I expect that you should be able to turn in the revised paper in less than 30 days, if not sooner. If your manuscript was reviewed, you will find the reviewers' comments below.

When submitting the revised version of your paper, please provide (1) point-by-point responses to the issues raised by the reviewers as file type "Response to Reviewers," not in your cover letter, and (2) a PDF file that indicates the changes from the original submission (by highlighting or underlining the changes) as file type "Marked Up Manuscript - For Review Only". Please use this link to submit your revised manuscript. Detailed instructions on submitting your revised paper are below.

Link Not Available

Sincerely,

Tomefa Asempa

Reviewer comments:

Reviewer #1 (Comments for the Author):

The authors responded correctly to all points I had raised in my initial review.

Reviewer #3 (Comments for the Author):

Excellent job making the suggested changes. I still have some concerns, primarily over the methodology of the study. This seems to be set-up as a cross-sectional, yet there are random comparisons between monomicrobial and polymicrobial cultures, acute and chronic infections, different infection sites, etc. that are not described as aims or objectives in the study background or methodology. Please provide clarity as to what exactly you're looking for and why there are so many different groups of comparisons, as this makes the data very challenging to interpret and find clinically meaningful impact.

Reviewer #4 (Comments for the Author):

This study attempts to describe the epidemiology of all osteoarticular infections at a single site in France and determine whether empiric antibiotic therapy following procedures can be deescalated prior to 14

days based on length to culture positivity.

Overall: There are grammatic, spelling, and punctuation errors that need to be corrected throughout.

Table 4: "Cefepim" should be "cefepime." Review document for misspelling elsewhere.

Line 89: Recommend "or linezolid" rather than "and linezolid."

Lines 106-108: As you don't set up a stewardship program within this study, consider wording this "provide data to inform a stewardship program . . ."

Line 183: Please clarify the relevance or significance of the underlying conditions.

Line 197: The spelling is Enterobacterales. Review for misspelling of this elsewhere.

Line 222: Table 4 discusses pip-taz and cefepime, not pip-taz and daptomycin. I assume "daptomycin" should read "cefepime"?

Preparing Revision Guidelines

Please return the manuscript within 60 days; if you cannot complete the modification within this time period, please contact me. If you do not wish to modify the manuscript and prefer to submit it to another journal, please notify me of your decision immediately so that the manuscript may be formally withdrawn from consideration by Microbiology Spectrum.

This study attempts to describe the epidemiology of all osteoarticular infections at a single site in France and determine whether empiric antibiotic therapy following procedures can be deescalated prior to 14 days based on length to culture positivity.

Overall: There are grammatic, spelling, and punctuation errors that need to be corrected throughout.

Table 4: "Cefepim" should be "cefepime." Review document for misspelling elsewhere.

Line 89: Recommend "or linezolid" rather than "and linezolid."

Lines 106-108: As you don't set up a stewardship program within this study, consider wording this "provide data to inform a stewardship program . . ."

Line 183: Please clarify the relevance or significance of the underlying conditions.

Line 197: The spelling is Enterobacterales. Review for misspelling of this elsewhere.

Line 222: Table 4 discusses pip-taz and cefepime, not pip-taz and daptomycin. I assume "daptomycin" should read "cefepime"?

Responses to reviewer's comments

Reviewer #1

The authors responded correctly to all points I had raised in my initial review.

Answer: we are thankful to the reviewer 1 for her/his positive comments and her/his initial remarks which helped us to improve the manuscript.

Reviewer #3

Excellent job making the suggested changes. I still have some concerns, primarily over the methodology of the study. This seems to be set-up as a cross-sectional, yet there are random comparisons between monomicrobial and polymicrobial cultures, acute and chronic infections, different infection sites, etc. that are not described as aims or objectives in the study background or methodology. Please provide clarity as to what exactly you're looking for and why there are so many different groups of comparisons, as this makes the data very challenging to interpret and find clinically meaningful impact.

Answer: we thank reviewer 3 for her/his very positive comments. As written in the manuscript, "the aim of this study was to describe the local OAI epidemiology with consideration of bacterial growth times to determine which antibiotic stewardship intervention could be implemented in case of negative culture after 2 days of incubation". However, we took the opportunity of the data collected to further analyzed our data by comparing monomicrobial and polymicrobial cultures, acute and chronic infections, different infection sites, etc. Therefore, these comparisons were not the objective but rather a collateral advantage of the work that have been done. The same kind of analysis were performed in similar papers (*e.g.* reference 7: Deroche L, Plouzeau C, Bémer P, Tandé D, Valentin AS, Jolivet-Gougeon A, Lemarié C, Bret L, Kempf M, Héry-Arnaud G, Corvec S, Burucoa C, Arvieux C, Bernard L; and the CRIOGO (Centre de Référence des Infections Ostéo-articulaires du Grand Ouest) Study Group. 2019. Probabilistic chemotherapy in knee and hip replacement infection: the place of linezolid. *Eur J Clin Microbiol Infect Dis.* 38:1659–1663).

Reviewer #4

Comments for the Author:

Thank you for the opportunity to review the revised manuscript "Local epidemiology of osteoarticular infections with consideration of bacterial growth times provides useful data to inform antimicrobial stewardship." The authors evaluated time to bacterial growth, and if this information could be used to guide changes to empirical antimicrobial therapy (EAT) as early as 2 days after surgical intervention. These results, particularly related to the timing of bacterial growth, are interesting and could be of use to other stewardship programs. The extensive revisions and response to reviewers is noted, and the authors' revisions largely address this reviewer's concerns sufficiently. The following comments are provided for further consideration.

Answer: we are thankful to the reviewer 4 for her/his positive comments and her/his initial remarks which helped us to improve the manuscript.

1. The abstract (line 52) still describes the study as a cohort study, as does the discussion (line 238). Additionally, I would suggest removing the term "prospective" as cross-sectional studies are non-directional (line 52, line 111).

Answer: as suggested, we removed the term “prospective” all along the manuscript.

2. One of the aims now focuses on establishing the right time to modify EAT in patients with negative cultures at 2 days. The final proposal suggests switch at 5 days based on the microbiology. Based on Table 2/2bis, the authors may wish to comment on the feasibility of changing pip/tazo to something without Pseudomonal coverage at day 2 or day 3, which would be consistent with the stated aim. This could be a preceding step on the proposed algorithm (Figure 2), still followed by stopping all gram negative coverage at day 5.

Answer: the reviewer 4 makes a very interesting comment. However, despite gram negative bacteria are much rare after day 2 or 3, they are no absent. This finding together to the resistance profile of these gram negative bacteria make the feasibility of changing pip/tazo to something without Pseudomonal coverage at day 2 or day 3 unlikely.

3. EAT with pip/tazo + daptomycin is very broad. It is no surprise that 90% of cases were adequately treated by this regimen. I still think that a comparison to a third generation cephalosporin would strengthen the conclusion that "EAT... is adapted to our local epidemiology", since the regimen of pip/tazo + daptomycin is likely to be effective >90% in most institutions. Pseudomonas was only 7% of all isolates, so a regimen without Pseudomonas coverage could arguably still be adequate >80% of the time. Exploring narrower spectrum empiric regimens is reasonable for an article concerned about stewardship. Similarly if the concern is maximizing the proportion of patients receiving active EAT, then comparison to meropenem or similar would also make sense. If the authors do not wish to do these evaluations (i.e. evaluate adequacy of alternative EAT regimens), then I would suggest also removing the comparison to cefepime and instead focus on the overall adequacy of the regimen at providing sufficient empiric coverage (90%).

Answer: we made the evaluation requested by reviewer 4. As seen in the updated table 4 here below, using imipenem instead of piperacillin-tazobactam would have increased the appropriateness of EAT for microbiologically proven OAI (n=144) to 94.5%, whereas if piperacillin-tazobactam would have been replaced by a third-generation cephalosporin, the appropriateness of EAT for microbiologically proven OAI would have decrease to 78%. Indeed, 8 OAI cases among the 144 microbiologically proven OAI were documented with at least one strain resistant to imipenem (*Pseudomonas aeruginosa* and/or *Stenotrophomonas maltophilia*); whereas 32 OAI cases among the 144 microbiologically proven OAI were documented with at least one strain resistant to a third-generation cephalosporin (anaerobes, ESBL producing Enterobacterales, *Pseudomonas aeruginosa*, *Stenotrophomonas maltophilia*, *Aeromonas hydrophila* and/or *Acinetobacter spp.*).

As already answered regarding carbapenem, the local policy restricts the usage of carbapenems to patients experiencing bacteremia due to a resistant *Enterobacterales* or known to have had OAI with a resistant bacterium in the preceding two month. Therefore, taken into account the only slight increase of the appropriateness of EAT for microbiologically proven OAI, we will not use imipenem.

We will neither recommend 3rd-generation cephalosporin since the decrease in the appropriateness of EAT for microbiologically proven OAI is too important comparing to the current recommendations in our setting.

We added some sentences about this in the “results” and “discussion” sections in the revised manuscript (lines 220-231 and 294-316 in the revised manuscript with revised marked highlighted).

4. Alternatively, consider modifying Table 4 to compare susceptibility of pathogens to the regimens (dapto + pip/tazo) vs (dapto + cefepime) instead of just the gram-negative drugs. This would also help clarify the discussion lines 287-290.

Answer: as previously answered to the first round of review, we did measure the susceptibility to daptomycin only for gram-positive bacteria resistant to piperacillin-tazobactam. Therefore, we do not have daptomycin susceptibility results for all the strains isolated in the frame of the study. Evaluating daptomycin susceptibility profile of all the strains isolated in our study was beyond the scope of the study. If needed for publication, we are willing to measure daptomycin MICs for the strains for which this data is missing; but no other reviewer requested that to be done.

5. The p-value on line 250 of the discussion does not match that of line 214 in the results. Please reconcile, or consider removing the statement in parentheses line 249-250 as it repeats information from the results.

Answer: as suggested, we removed the statement in parentheses line 249.

6. Line 123, I believe facia should be facial

Answer: we apologize for this typing error that was corrected in the revised manuscript.

Reviewer #5

This study attempts to describe the epidemiology of all osteoarticular infections at a single site in France and determine whether empiric antibiotic therapy following procedures can be deescalated prior to 14 days based on length to culture positivity.

Overall: There are grammatic, spelling, and punctuation errors that need to be corrected throughout.

Table 4: "Cefepim" should be "cefepime." Review document for misspelling elsewhere.

Answer: we apologize for the typing errors that were corrected in the revised manuscript.

Line 89: Recommend "or linezolid" rather than "and linezolid."

Lines 106-108: As you don't set up a stewardship program within this study, consider wording this "provide data to inform a stewardship program . . ."

Answer: the modifications suggested by reviewer 5 at lines 89 and line 106 have been made.

Line 183: Please clarify the relevance or significance of the underlying conditions.

Answer: the underlying conditions quoted are risk factors for OAI; we added a reference to support the sentence.

Line 197: The spelling is Enterobacterales. Review for misspelling of this elsewhere.

Answer: we apologize for the typing errors that were corrected in the revised manuscript.

Line 222: Table 4 discusses pip-taz and cefepime, not pip-taz and daptomycin. I assume "daptomycin" should read "cefepime"?

Answer: we modified the whole paragraph in order to answer to reviewer 4' question 3.

October 21, 2022

Dr. Alexandra Aubry
AP-HP
Sorbonne Université
91, boulevard de l'hôpital
Paris cedex 13 75013
France

Re: Spectrum01430-22R2 (Local epidemiology of osteoarticular infections with consideration of bacterial growth times provides useful data to inform antibiotic stewardship)

Dear Dr. Alexandra Aubry:

Appreciate edits to paper after valuable comments from our reviewers. Your manuscript has been accepted, and I am forwarding it to the ASM Journals Department for publication. You will be notified when your proofs are ready to be viewed.

Sincerely,

Tomefa Asempa
Editor, Microbiology Spectrum
